# A scalable and modular automated pipeline for stitching of large electron microscopy datasets

**Gayathri Mahalingam[1]\*[†], Russel Torres[1][†], Daniel Kapner[1], Eric T Trautman[2], Tim Fliss[1], Shamishtaa Seshamani[1], Eric Perlman[3], Rob Young[1], Samuel Kinn[1], JoAnn Buchanan[1], Marc M Takeno[1], Wenjing Yin[1], Daniel J Bumbarger[1], Ryder P Gwinn[4], Julie Nyhus[1], Ed Lein[1], Steven J Smith[1], R Clay Reid[1], Khaled A Khairy[5], Stephan Saalfeld[2], Forrest Collman[1], Nuno Macarico da Costa[1]\***

[1]Allen Institute for Brain Science, Seattle, United States; [2]HHMI Janelia Research Campus, Ashburn, United States; [3]Yikes LLC, Baltimore, United States; [4]Epilepsy Surgery and Functional Neurosurgery, Swedish Neuroscience Institute, Seattle, United States; [5]St. Jude Children's Research Hospital, Memphis, United States

**\*For correspondence:**
gayathrim@alleninstitute.org
(GM);
nunod@alleninstitute.org
(NMdC)

[†]These authors contributed
equally to this work

**Competing interest:** See page
24

**Reviewing Editor:** Albert
Cardona, University of
Cambridge, United Kingdom

**Abstract** Serial-section electron microscopy (ssEM) is the method of choice for studying macroscopic biological samples at extremely high resolution in three dimensions. In the nervous system, nanometer-scale images are necessary to reconstruct dense neural wiring diagrams in the brain, so -called *connectomes*. The data that can comprise of up to $10^8$ individual EM images must be assembled into a volume, requiring seamless 2D registration from physical section followed by 3D alignment of the stitched sections. The high throughput of ssEM necessitates 2D stitching to be done at the pace of imaging, which currently produces tens of terabytes per day. To achieve this, we present a modular volume assembly software pipeline *ASAP* (Assembly Stitching and Alignment Pipeline) that is scalable to datasets containing petabytes of data and parallelized to work in a distributed computational environment. The pipeline is built on top of the *Render* Trautman and Saalfeld (2019) services used in the volume assembly of the brain of adult *Drosophila melanogaster* (Zheng et al. 2018). It achieves high throughput by operating only on image meta-data and transformations. ASAP is modular, allowing for easy incorporation of new algorithms without significant changes in the workflow. The entire software pipeline includes a complete set of tools for stitching, automated quality control, 3D section alignment, and final rendering of the assembled volume to disk. ASAP has been deployed for continuous stitching of several large-scale datasets of the mouse visual cortex and human brain samples including one cubic millimeter of mouse visual cortex (Yin et al. 2020); Microns Consortium et al. (2021) at speeds that exceed imaging. The pipeline also has multi-channel processing capabilities and can be applied to fluorescence and multi-modal datasets like array tomography.

## Editor's evaluation

Datasets in volume electron microscopy have been growing fruit of the labor of the combined efforts of sample preparation specialists and electron microscopy engineers. A missing piece has been a method for the automation of the composition of continuous volumes out of collections of individual image tiles capable of handling the growing scales of the datasets. Pushing the boundaries of what is possible, this work illustrates how a successful approach looks like, demonstrated by its application to cubic millimeter volumes imaged at nanometer resolution. All being said, this work is but step 1 of a two-step process, whereby first a coarse but mostly correct alignment is computed, and then a refinement step using more local cues and with existing methods is applied, setting the

stage for the subsequent automated reconstruction of neuronal arbors and their synapses from which to infer a cellular connectome.

## Introduction

Serial section electron microscopy (ssEM) provides the high spatial resolution in the range of a few nanometers per pixel that is necessary to reconstruct the structure of neurons and their connectivity. However, imaging at a high resolution produces a massive amount of image data even for a volume that spans a few millimeters. For example, a cubic millimeter of cortical tissue imaged at a resolution of $4 \times 4 \times 40$ nm$^3$ generates more than a petabyte of data and contains more than 100 million individual image tiles (W et al. 2020). These millions of images are then stitched in 2D for each section and aligned in 3D to assemble a volume that is then used for neuronal reconstruction. With parallelized high-throughput microscopes producing tens of terabytes of data per day, it is necessary that this volume assembly process is automated and streamlined into a pipeline, so that it does not become a bottleneck. The ideal pipeline should be capable of processing data at the speed of imaging (*Lichtman et al., 2014*) and produce a high-fidelity assembled volume. To match the speed of the EM imaging, the volume assembly pipeline needs to be automated to handle millions of images per day from multiple microscopes. Though electron microscopy is notorious for creating very large datasets, other volume microscopy technologies that collect 3D data would also gain from advances in automated and scalable methods for stitching and alignment.

Imaging and 3D reconstruction of biological samples usually involve a series of stages from preparing the tissue, cutting it into serial sections, imaging them using an image acquisition system, 2D registration and 3D alignment of those serial sections, and finally 3D segmentation (*Figure 1a*). Each serial section imaged comprises of several hundreds to several thousands of images depending on the resolution at which they are imaged. The volume assembly process that registers and aligns these images works under the assumption that the images within a serial section carry some overlap between neighboring tile images (*Figure 1b and c*). The images are registered based on some points of interest that are extracted from the overlapping region (*Figure 1d*). This also requires the raw tile images to be corrected for any lens distortion effects that arise from the acquisition system (*Figure 1e–g*). The stitched serial sections can then be 3D aligned using a similar process of matching patterns between the montages. The challenge in the volume assembly process arises when the throughput has to be matched with the acquisition system for large-scale datasets. Also, a highly accurately aligned 3D volume is necessary for further segmentation and reconstruction.

Several tools used in various stages of volume assembly pipelines perform image registration by extracting and matching similar features across overlapping images (*Cardona et al., 2012*; *Wetzel et al., 2016*; *Bock et al., 2011*; *Karsh, 2016*). Image registration using Fourier transformation (*Wetzel et al., 2016*) was used to successfully align mouse and zebrafish brain datasets acquired using wager mapper ssEM imaging technology. The Fiji (*Schindelin et al., 2012*; *Rasband, 2012*) plugin TrakEM2 (*Cardona et al., 2012*) includes a comprehensive set of tools and algorithms to perform stitching and alignment of various types of microscopy image formats. AlignTK (*Bock et al., 2011*) implements scalable deformable 2D stitching and serial section alignment for large serial section datasets using local cross-correlation. An end-to-end pipeline to perform volume assembly and segmentation using existing tools was developed by R. *Vescovi, 2020* and was designed to run on varied computational systems. The pipeline was shown to process smaller datasets through supercomputers efficiently. While these approaches have been successfully used in the volume assembly of smaller datasets, they do not scale well for large-scale datasets, lack support for different classes of geometric transformations, or do not incorporate reliable filters for false matches due to imaging artifacts (*Khairy et al., 2018*).

We propose a volume assembly pipeline – *ASAP* (Assembly Stitching and Alignment Pipeline; https://github.com/AllenInstitute/asap-modules; *Mahalingam, 2022*) – that is capable of processing petascale EM datasets with high-fidelity stitching and processing rates that match the speed of imaging. Our pipeline is based on the volume assembly framework proposed in *Zheng et al., 2018* and is capable of achieving high throughput by means of metadata operations on every image in the dataset. The metadata and transformations associated with each image are stored in a MongoDB database fronted by *Render* (*Trautman and Saalfeld, 2019*) services to dynamically render the output

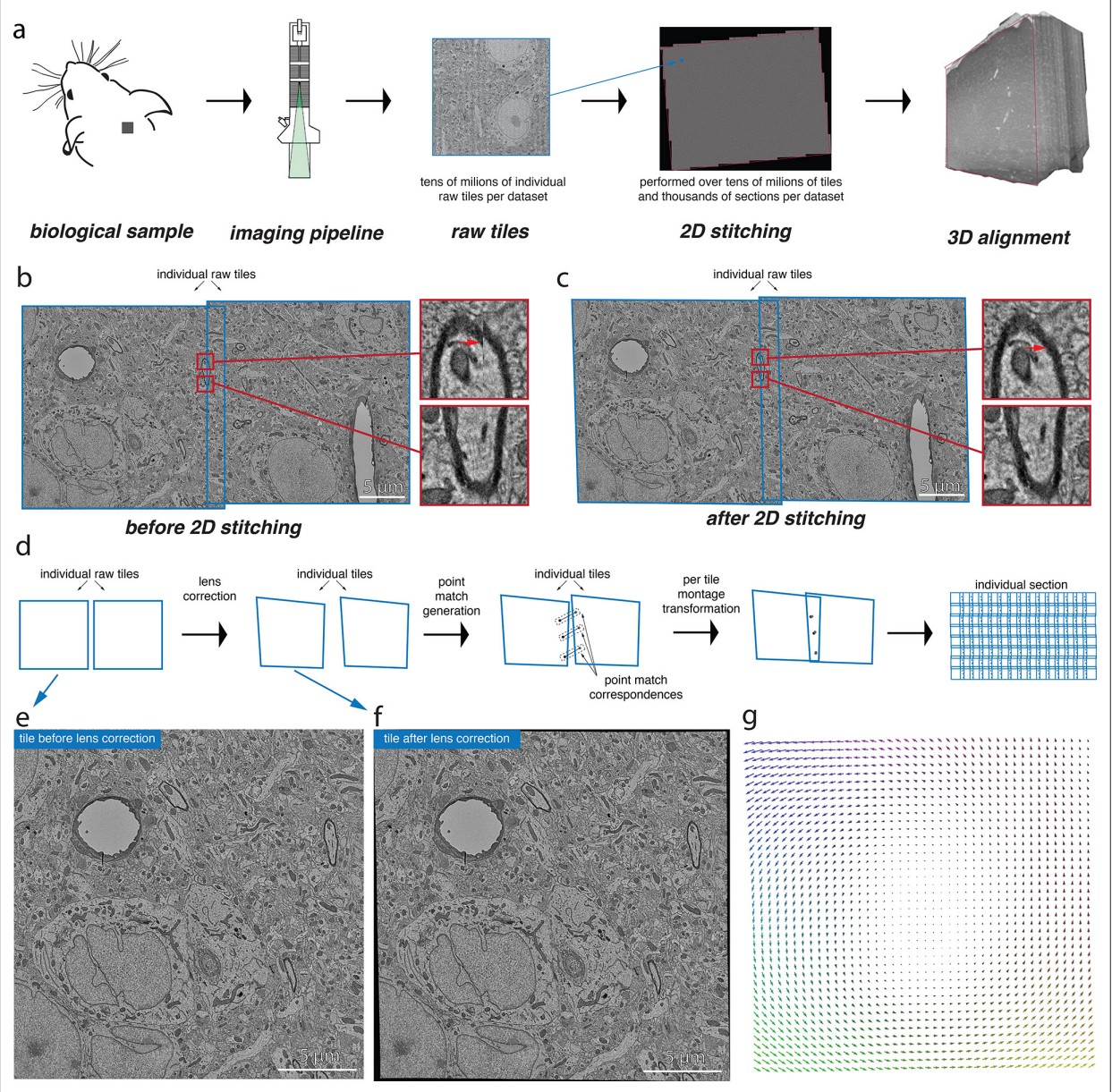

**Figure 1.** Volume assembly pipeline. (**a**) Different stages of the electron microscopy (EM) dataset collection pipeline. The biological sample is prepared and cut into thin slices that are imaged using the desired image acquisition system (electron microscopy for datasets discussed in this work). The raw tile images from each section are then stitched together in 2D followed by a 3D alignment of them. (**b**) A pair of raw tile images before 2D stitching. The tiles have a certain overlap between them and are not aligned (the zoomed-in regions show the misalignment) and hence require a per-tile transformation to *stitch* them together. (**c**) The pair of tile images from (**b**) after stitching is performed. The zoomed-in regions illustrate the alignment of these images after stitching. (**d**) Conceptual diagram illustrating the series of steps that are involved in the 2D stitching of the serial sections. The steps include computation of lens distortion correction transformation followed by generation of point correspondences between the overlapping tile images and, finally, computation of per-tile montage transformations using the point correspondences. (**e**) A raw tile image without any lens distortion correction. (**f**) Tile image from (**e**) after lens distortion correction transformation is applied. (**g**) A quiver plot showing the magnitude and direction of distortion caused by the lens from the acquisition system.

at any stage in the pipeline. The effectiveness of the pipeline has been demonstrated in the volume assembly of multiple petascale volumes and integrates well with SEAMLeSS (*Macrina et al., 2021*), which provided the final 3D alignment of these volumes.

The pipeline described here for assembly of large connectomics volumes is divided into two sections: (1) a software package that is scalable, modular, and parallelized and is deployable in varied computing environments to perform volume assembly of EM serial sections; (2) a workflow engine and

a volume assembly workflow that utilizes these tools to automate the processing of raw EM images from a multiscope setup using high-performance computing (HPC) systems. The tools in ASAP are open source and include abstract-level functionalities to execute macro-level operations that execute a series of steps required for the execution of different stages of the pipeline. An example of such a macro operation is the computation of point-match correspondences, which requires the generation of tile pairs and generation of point matches using those tile pairs. The modularity of the tools allows for easy implementation of other algorithms into the pipeline without making major changes to the existing setup. The software tools can be easily deployed in different computing environments such as HPC systems, cloud-based services, or on a desktop computer in a production-level setting. The software stack also includes a set of quality c ontrol (QC) tools that can be run in an automated fashion to assess the quality of the stitched montages. These software tools can be easily utilized by workflow managers running the volume assembly workflow to achieve high throughput. The tools are designed to generalize well for other datasets from different domains (that carry the assumption of generating overlapping images) and can be adapted to process such datasets. We have also developed a work-flow manager *BlueSky* (https://github.com/AllenInstitute/blue_sky_workflow_engine; *Melchor et al., 2021*) that implements the volume assembly workflow using our software stack. The proposed pipe-line combined with *BlueSky* has been successfully used to stitch and align several high-resolution mm³ EM volume from the mouse visual cortex and a human dataset at speeds higher than the imaging rate of these serial sections from a highly parallelized multiscope setup.

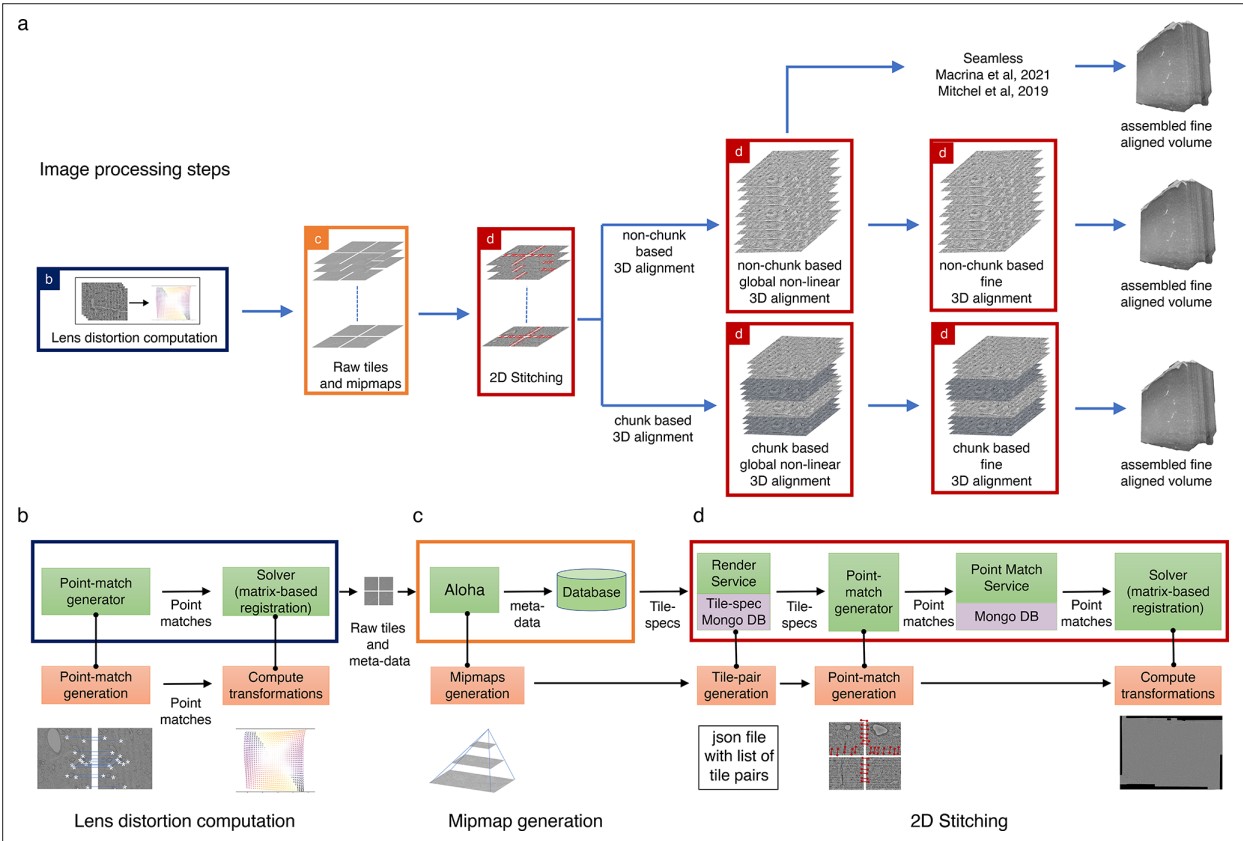

**Figure 2.** Assembly Stitching and Alignment Pipeline (ASAP) – volume assembly workflow. (**a**) The different steps of image processing in *ASAP* for electron microscopy (EM) serial sections. The infrastructure permits multiple possible strategies for 3D alignment, including a chunk-based approach in case it is not possible to 3D align the complete dataset at once, as well as using other workflows outside ASAP (*Macrina et al., 2021*; https://www.microns-explorer.org/cortical-mm3) for fine 3D alignment with the global 3D aligned volume obtained using ASAP. (**b–d**) Representation of different modules in the software infrastructure. The green boxes represent software components, the orange boxes represent processes, and the purple processes represent databases. The color of the outline of the box matches its representation in the image processing steps shown in (**a**). (**b**) Schematic showing the lens distortion computation. (**c**) Schematic describing the process of data transfer and storage along with MIPmaps generation using the data transfer service *Aloha*. (**d**) Schematic illustrating the montaging process of serial sections. The same software infrastructure of (**d**) is then also used for 3D alignment as shown by the red boxes in (**a**).

# Results

## Development of a stitching and alignment pipeline

The pipeline (ASAP) described in this work is based on the principles described by *Kaynig et al., 2010*, *Saalfeld et al., 2010*, and *Zheng et al., 2018*, and scales the software infrastructure to stitch and align petascale datasets. It includes the following stages: (1) lens distortion computation, (2) 2D stitching, (3) global section-based nonlinear 3D alignment, (4) fine 3D alignment, and (5) volume assembly. ASAP performs feature-based stitching and alignment in which point correspondences between two overlapping images are extracted and a geometric transformation is computed using these point correspondences to align the images.

*Figure 2* shows the volume assembly pipeline (ASAP) for building 3D reconstruction out of serial section transmission electron microscopy (ssTEM) images. First, single images from serial sections from ssTEM are collected. As the field of view is limited, multiple images that overlap with each other are imaged to cover the entire section. Images acquired by ssTEMs can include dynamic nonlinear distortions brought about by the lens system. A compensating 2D thin plate spline transformation is derived using a custom triangular mesh-based strategy (*Collins et al., 2019*) based on point correspondences of overlapping image tiles as in *Kaynig et al., 2010*. The point correspondences (also referred to as point matches) are extracted using SIFT *Lowe, 2004* and a robust geometric consistency filter using a local optimization variant of RANSAC *Fischler and Bolles, 1981* and robust regression (*Saalfeld et al., 2010*) (see 'Methods' for more details). These point correspondences, in lens-corrected coordinates, are then used to find a per-image affine/polynomial transformation that aligns the images in a section with each other to create a *montage*. The affine/polynomial transformations are computed using a custom Python package, BigFeta, which implements a direct global sparse matrix solving strategy based on *Khairy et al., 2018*. The stitched montages are then globally aligned with each other in 3D. The 3D global alignment is performed by extracting point correspondences between low-resolution version of the 2D stitched sections and solved with BigFeta to obtain a thin plate spline per section transformation. This 3D alignment is the result of a progressive sequence of rotational, affine, and thin plate spline solves with tuned regularization parameters such that each solution initializes the next more deformable, yet increasingly regularized transformation. The globally aligned transformations can then be used as an initialization for computing finer and denser alignment transformations (an example of this is the fine alignment described in *Macrina et al., 2021*), which is computed on a per-image basis at a much higher resolution. Several iterations of the global 3D alignment are performed to achieve a good initialization for the fine alignment process. For all the datasets presented in this article, the 2D stitching and global alignment was performed using ASAP, and afterward the data was materialized and transferred outside of ASAP for fine alignment using SEAMLeSS (*Macrina et al., 2021*).

In a continuous processing workflow scenario, the serial sections from multiple ssTEMs are stitched immediately once they are imaged. 3D alignment is performed on chunks of contiguous sections that partially overlap with their neighboring chunks. These independently 3D aligned chunks can be assembled to a full volume by aligning them rigidly and interpolating the transformations in the overlapping region (*Figure 2*).

## Software infrastructure supporting stitching and alignment

Our software infrastructure is designed to support EM imaging pipelines such as piTEAM (*Wetzel et al., 2016*) that produce multiple serial sections from a parallelized scope setup every hour. The infrastructure is designed for processing petascale datasets consisting of millions of partially overlapping EM images. The infrastructure consists of four core components: (1) a modular set of software tools that implements each stage of ASAP (asap-modules), (2) a service with REST APIs to transfer data from the microscopes to storage hardware (Aloha); (3) REST APIs for creating, accessing, and modifying image metadata (Render); and (4) a matrix-based registration system (BigFeta). Below we provide a brief description of these components with a more detailed description in the section 'ASAP modules'.

ASAP is implemented as a modular set of tools that includes abstract-level functions to execute for each stage of the volume assembly pipeline. It also includes QC tools to assess stitching quality, render results to disk at any stage of the pipeline, obtain optimal parameters for computing point correspondences, and obtain optimal parameters for solving optimal transformations. Asap-modules

is supported by *render-python* for read/writes to the database and *argschema* for its input and output data validation (see 'Methods' for more details).

Aloha is an image transfer service (*Figure 2c*) that receives raw images and their metadata from the microscopes, stores them in primary data storage, and losslessly compresses the original data to reduce the storage footprint. It includes REST APIs for clients to GET/POST images and their metadata. It also produces downsampled representations of the images for faster processing and visualization.

Render (*Trautman and Saalfeld, 2019*) provides logic for image transformation, interpolation, and rendering. It is backed by a MongoDB document store that contains JSON (JavaScript Object Notation) tile specifications with image metadata and transformations. Render's REST APIs are accessed by asap-modules using render-python to create, access, and modify image metadata in the database. The REST APIs allow the user to access the current state of any given set of image tiles during the stitching process. Render also includes a point-match service that handles the storage and retrieval of point correspondences in a database since computing point correspondences between millions of pairs of images are computationally expensive. Another advantage of storing the point correspondences in a database is that it is agnostic to the algorithm that is used for the computation of these point correspondences. The *point-match service* (*Figure 2c and e*) handles the data ingestion and retrieval from the database using REST APIs with both operations being potentially massively distributed.

BigFeta is a matrix-based registration system that estimates the image transformations using the point correspondences associated with the image. BigFeta includes transformations such as rotations to implement rigid alignments, and 2D thin plate spline transformations that are useful for 3D image alignments. BigFeta can also be integrated with distributed solver packages such as PETSc (*Balay et al., 2019*) for solving large sparse matrices involving billions of point correspondences.

We also developed a workflow manager *BlueSky* as well as an associated volume assembly workflow to automatically process serial sections as they are continuously ingested during the imaging process. It utilizes the abstract-level functions in asap-modules to create workflows for each stage of the volume assembly pipeline.

Our alignment pipelines operate only on metadata (point correspondences and transformations) derived from image tiles – a feature derived from the Render services, thus allowing efficient processing of petascale datasets and the feasibility of real-time stitching with proper infrastructure. Where possible, the pipeline works with downscaled versions of image tiles (MIPmaps) that dramatically increases processing speed and reduces disk usage as raw data can be moved to a long-term storage for later retrieval.

Beyond the use of this software infrastructure for EM data, which drove the development that we describe in this article, the pipeline also has multichannel processing capabilities and can be applied to fluorescence and multimodal datasets like array tomography (see Figure 8).

## Data acquisition and initiation of image processing

An important first step in our pipeline is the correction of lens distortion effects on raw images. Lens distortions are calculated from a special set of images with high tile overlap. These *calibration montages* are collected at least daily and after any event that might affect the stability of the beam (e.g., filament replacement). This step is followed by the acquisition of the neuroanatomical dataset, for which a bounding box is drawn around the region of interest (ROI) in each ultra-thin section. In certain situations, multiple ROIs are required per section. The volume assembly workflow accepts multiple entries referencing the same placeholder label to support reimaging. At the end of each acquisition session, the tiles, tile manifest, and session log are uploaded to the data center storage cluster and the lens correction and montaging workflows in the volume assembly workflow are triggered. *Figure 3* shows the specialized services that facilitate data transfer and tracking from high-throughput microscopes to shared compute resources.

This infrastructure was used to process multiple petascale datasets, including a 1 mm$^3$ (mouse dataset 1) of the mouse brain that is publicly available at microns-explorer (*MICrONS Consortium et al., 2021*). Over 26,500 sections were imaged at 4 nm/pixel resolution using five microscopes, running in a continuous and automated fashion (W et al. 2020). Each montage is composed of ~5000 tiles of 15 µm × 15 µm with an overlap of 13% in both $x$ and $y$ directions. The total file size of a single montage is about 80 GB, and thus a daily throughput of 3.6 TB per system is produced in a continuous

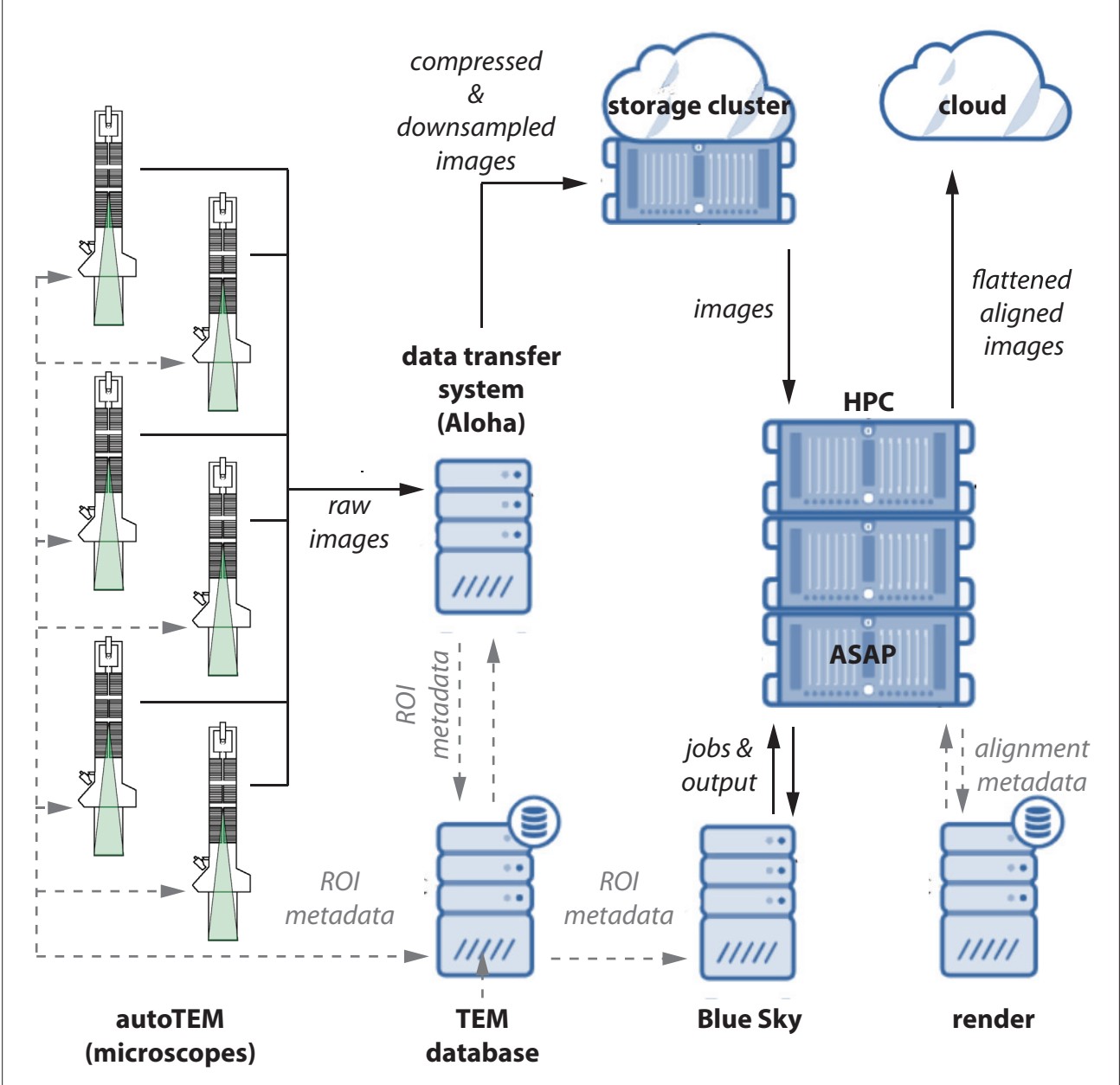

**Figure 3.** Data flow diagram. A schematic diagram showing the flow of image data, metadata, and processed data between microscopes. Raw images and metadata are transferred from microscopes to our data transfer system (*Aloha*) and transmission electron microscopy (TEM) database, respectively. Aloha generates MIPmaps and compresses images and transfers them to the storage cluster for further processing by *ASAP*. Metadata is transferred to BlueSky through TEM database, which triggers the stitching and alignment process. The metadata from the stitching process is saved in the Render services database. The final assembled volume is transferred to the cloud for further fine alignment and segmentation. The hardware configurations are presented in Appendix 5.

imaging scenario. Part of the dataset was imaged using a 50 MP camera with an increased tile size to 5408 × 5408 pixels. This resulted in montages with ~2600 tiles at an overlap of 9% in both $x$ and $y$ directions. The infrastructure was also used to process two other large mouse datasets and a human dataset. The details about these datasets are shown in *Table 1*, where the ROI size and total nonoverlapping dataset size (without repeated pixels) were determined from montage metadata, including pixel size and nominal overlap.

**Table 1.** Details of datasets processed using Assembly Stitching and Alignment Pipeline (ASAP) – volume assembly pipeline.

| Dataset | No. of sections | Pixel resolution (nm/pixel) | Image size | Tile overlap | Total size (PiB) | Total size (nonoverlap) (PiB) | ROI size (in microns) |
|---------|------------------|------------------------------|-------------|---------------|-------------------|-------------------------------|------------------------|
| Mouse dataset 1 | 19,945 | 3.95–4 | $3840 \times 3840$ (20MP camera), $5408 \times 5408$ (50MP camera) | 13%, 9% | 1.6 | 1.2 | $1300 \times 870$ |
| Mouse dataset 2 | 17,593 | 4.67–5 (4.78 mean) | $5376 \times 5376$ | 9–10%, 9.4% mean | 0.984 | 0.807 | $1191 \times 815$ |
| Mouse dataset 3 | 17,310 | 3.8–4.15 (3.91 mean) | $5376 \times 5376$ | 9–10%, 9.1% mean | 0.665 | 0.549 | $647 \times 672$ |
| Human | 9673 | 4.65–4.95 (4.78 mean) | $5376 \times 5376$ | 9–10%, 9.5% mean | 1.18 | 0.968 | $2315 \times 987$ |

ROI, region of interest.

## Automated petascale stitching

Besides stitching and aligning large-scale datasets, a requirement for the volume assembly pipeline is to achieve a rate that matches or exceeds the imaging speed so as to provide rapid feedback on issues with the raw data encountered during the stitching process. This is achieved in our pipeline using an automated workflow manager (BlueSky) that executes the volume assembly pipeline to continuously process serial sections from five different autoTEMs (*Wetzel et al., 2016*).

The images from the autoTEMs are transferred to the Aloha service without sending them to storage servers directly. The Aloha service generates MIPmaps, compresses the raw images, and then writes them to the storage servers. The sections processed by Aloha are then POSTed to the BlueSky workflow manager, which initiates the montaging process. During an imaging run, each microscope uploads raw data and metadata to Aloha using a concurrent upload client. Limitations of the autoTEM acquisition computers cap the Aloha client throughput at 0.8–1.2 Gbps per microscope, which is sufficient for daily imaging with a 50 MP camera as described in *Yin et al., 2020*; *Wetzel et al., 2016*. Transferring previously imaged directories from high-performance storage servers has shown that an Aloha deployment on multiple machines is capable of saturating a 10 Gbps network uplink. The serial sections are assigned *pseudo z* indices to account for errors in metadata from the scopes such as barcode reading errors that assigns incorrect z indices. The lens correction workflow is triggered to compute a transformation that can correct lens distortion effects on the raw images. This transformation is updated in the image metadata so as to be used in subsequent stages of volume assembly. The montaging workflow in BlueSky triggers the generation of point correspondences and stores them in the database using the point-match service, followed by calculating the globally optimal affine/polynomial transformation for each image tile in the montage using the BigFeta solver. The transformations are saved as metadata associated with each tile image in the Render services database. The montages go through an automated QC process to ensure a high-fidelity stitching (see 'Automated montage QC'), followed by a global 3D alignment of the entire dataset.

ASAP is capable of performing the global 3D alignment in chunks, making it scalable to use in larger datasets or with fewer computational resources. However, all our datasets have been 3D aligned as a single chunk. The montages are rendered to disk at a scale of 0.01 and point correspondences are computed between the neighboring sections represented by their downsampled versions. A per-section thin plate spline transformation is computed using 25–49 control points in a rectangular grid. The per-section transformation is then applied to all the tile images in that section to globally align them in 3D.

## Automated montage QC

QC is a crucial step at each stage of processing in EM volume assembly to ensure that the outcome at each stage is of high quality. ASAP-modules include a comprehensive set of tools to perform QC of the computed lens correction transformations, stitched montages, and 3D aligned volume. These tools are integrated within the lens correction and montaging workflow in the volume assembly workflow to automatically compute statistical metrics indicating the stitching quality and also generates maps of montages showing potential stitching issues (see *Figure 4*). The stitched montages that pass

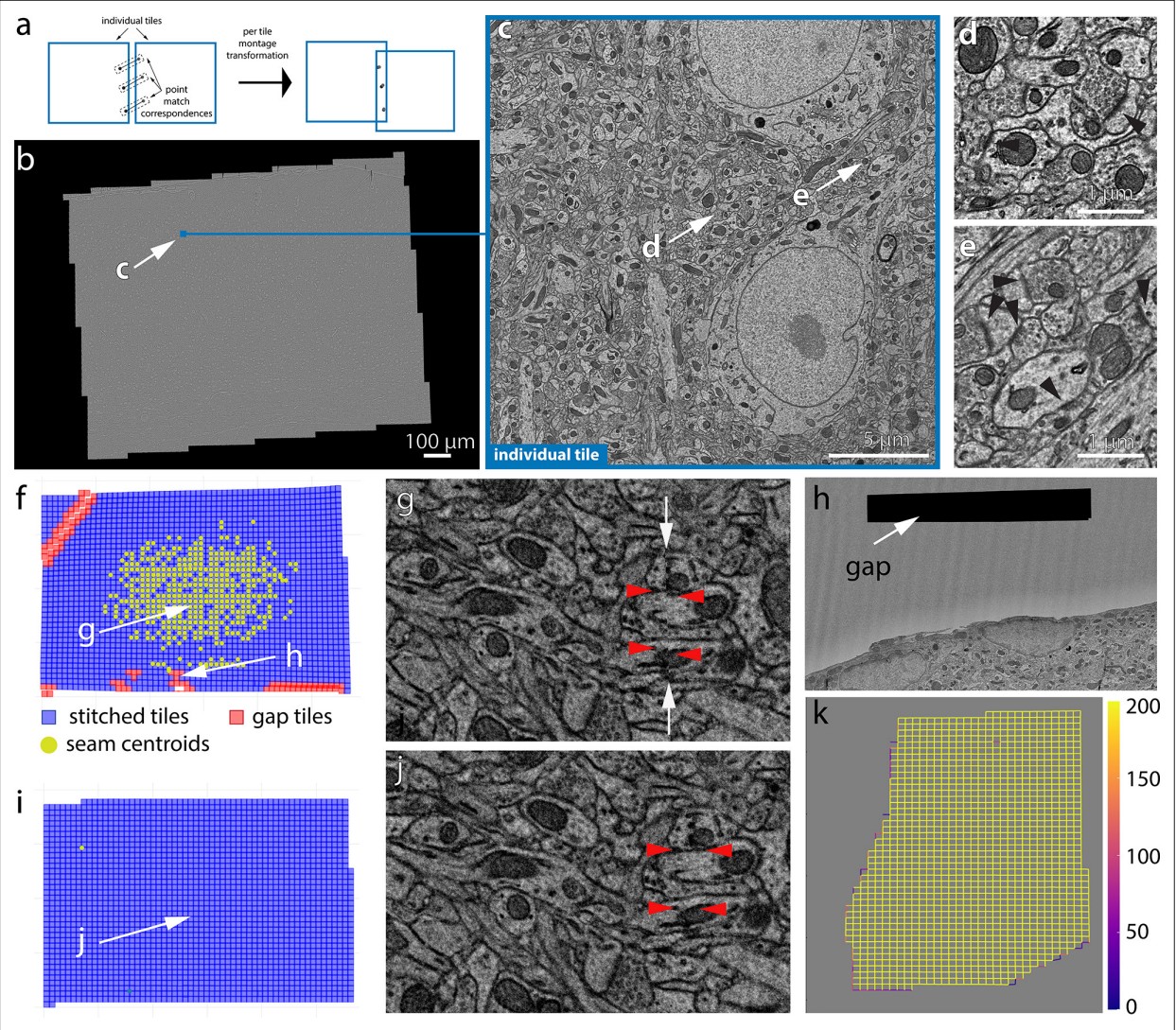

**Figure 4.** 2D stitching and automated assessment of montage quality. (**a**) Schematic diagram of the montage transformation using point correspondences. (**b**) Montage 2D stitched section from mouse dataset 1 (publicly available at https://www.microns-explorer.org; **MICrONS Consortium et al., 2021**). (**c**) Single-acquisition tile from the section in (**b**). (**d, e**) Detail of synapses (arrow heads) from the tile shown in (**c**). (**f**) Quality control (QC) plot of a stitched electron microscopy (EM) serial section with nonoptimal parameters. Each blue square represents a tile image of how they appear aligned in the montage. The red squares represent tile images that have gaps in stitching with neighboring tile images and are usually located in regions with resin or film. (**g**) A zoom-in region of the 2D montage in (**f**) showing the seam (white arrows) between tiles causing misalignment (red arrowheads) between membranes. (**h**) A zoomed-in region of the section showing a tile having a gap with its neighbors. (**i**) QC plot of a stitched EM serial section after parameter optimization. (**j**) A zoom-in region of the 2D montage in (**i**) showing no seams in the same region as in (**g**). The red arrowheads show the same locations as in (**g**). (**k**) A schematic plot representing the number of point correspondences between every pair of tile images for a section of the human dataset. Each edge of the squares in the plot represents the existence of point correspondences between tile images centered at the end points of the edge. The color of the edge represents the number of point correspondences computed between those tile image pairs.

QC are automatically moved to the next stage of processing, thus enabling faster processing with minimal human intervention but ensuring a high-quality volume assembly.

The stitching issues that are identified include misalignments between stitched tiles, gaps in montages, and seams. These issues are identified based on the mean residual between point correspondences from every pair of tiles. This represents how well the point correspondences have aligned from each of the tiles after montage transformations are applied to them (**Figure 4a**). This metric is represented in pixel distance and is used to locate areas of misalignments and seams. The gaps in stitching are identified by means of how many neighbor a tile image has before and after stitching and

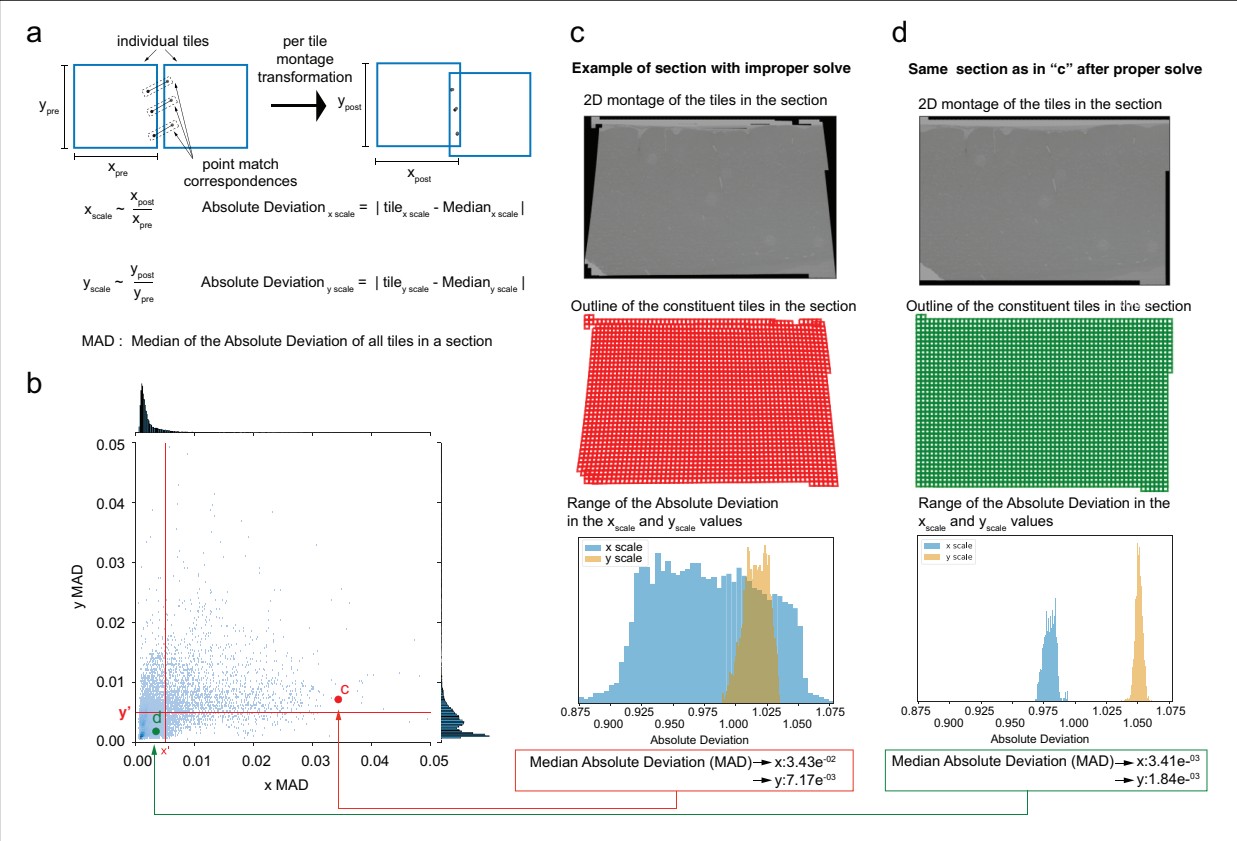

**Figure 5.** Median absolute deviation (MAD) statistics for montage distortion detection. (**a**) Schematic description of computation of MAD statistics for a montage. (**b**) A scatter plot of $x$ and $y$ MAD values for each montage. A good stitched section without distorted tile images falls in the third quadrant (where point d is shown). (**c**) An example of a distorted montage of a section solved using unoptimized set of parameters. Row 1 shows the downsampled version of the montaged section, row 2 shows the quality control (QC) plot of the section showing the distortions, and row 3 shows the $x$ and $y$ absolute deviation distribution for the unoptimized montage. (**d**) Section shown in (**c**) solved with optimized parameters with row 1 showing the downsampled montage, row 2 showing the QC plot of the section, and row 3 showing the $x$ and $y$ absolute deviation distribution for the section.

The online version of this article includes the following figure supplement(s) for figure 5:

**Figure supplement 1.** Parameter optimization.

based on their area of overlap with its neighbors. A seam appears as misalignment between a tile and many of its neighbors and is identified using a cluster of point correspondences whose residuals are above a certain threshold. In addition to these metrics, we also compute the mean absolute deviation (MAD) (*Figure 5*) that measures the amount of distortion a tile image undergoes with the transformation. The MAD statistics is a measure using which we identify montages that are distorted (*Figure 5*) once it passes the automated QC identifying other issues. Since the crux of our computations is based on the point correspondences, we also generate plots to quickly visualize the density of point correspondences between tile images within a section (*Figure 4k*).

The QC maps (*Figure 4f and i*) of the montages provide a rapid means to visually inspect and identify stitching issues associated with the montage without the need to materialize or visualize large-scale serial sections. The QC map reveals the location of gaps and seams between tiles in addition to providing an accurate thumbnail representation of the stitched section. The QC maps also provide an interactive way for the user to click on the thumbnail representation of a tile to visualize the tile image along with its neighbors in the stitched montage. This provides a means to quickly inspect individual tiles that have stitching issues. While the QC maps provide a quick view of the issues related to a montage, the Neuroglancer (*Neuroglancer, 2010*) tool can further facilitate the dynamic rendering of an ROI or the entire montaged section for further inspection. This provides the advantage of not requiring to render the intermediate output to disk.

A seam (*Figure 4g*) is defined as a misalignment between two tiles and is identified by means of the pixel residuals between point correspondences between the tiles. Misalignments can be eliminated by solving correct transformations using optimized sets of parameters. A gap between tile images (*Figure 4h*) is usually the result of inaccurate montage transformations that are caused by lack of point correspondences between tile pairs where the gap appears. Tile pairs that include features like blood vessel, resin, or film region, etc. (see *Figure 4h*), lack point correspondences, thus causing a gap between the tiles during stitching. The stitching issues associated with the resin or film region are ignored, while the gaps in tiles containing blood vessels are solved with optimal parameters to ensure no misalignments between the tile and its neighbors. The tile images that are entirely part of a blood vessel lack textural features for generation of point matches and hence are dropped by the solver during montaging. However, tile images that partially cover the blood vessel region undergo generation of point correspondences at a higher resolution followed by montaging using optimal parameters. This usually resolves the misalignments, but our framework does not limit the use of other algorithms such as phase correlation or cross-correlation for resolving such issues.

Sections that failed QC are examined by a human proofreader and moved to the appropriate stage of reprocessing. A manual proofreading process typically includes examining the QC plot for issues and further visualizing those areas in montage with issues to ensure that those issues either correspond to resin or film region tiles or tiles corresponding to tissue region. The regions with misalignments are further examined to send them to the appropriate stage of processing. If the misalignments are caused due to insufficient point correspondences, then they are sent to the point-matching stage of the montage workflow for generation of point correspondences at a higher resolution. Misaligned sections with sufficient point correspondences are sent to the solver stage with new parameters. These parameters were heuristically chosen by means of a parameter optimization algorithm based on the stitching quality metrics (see 'Montage parameter optimization' for more details and *Figure 5— figure supplement 1* for optimized parameter selection plots).

Unoptimized parameters can also lead to distorted montages where individual tiles are distorted (see *Figure 5c and d* for distorted and undistorted versions of the same montage). The median absolute deviation (MAD) (*Figure 5a and b*) statistic provides a computational assessment of the quality of the montage and aids in the selection of optimized set of parameters to solve for a montage. The optimal $x$ and $y$ MAD statistic values were heuristically selected for every dataset.

## Performance of the volume assembly pipeline: ASAP

High-quality 2D stitching and 3D alignment are necessary for accurate neuroanatomy reconstruction and detection of synaptic contacts. The 2D stitching quality is assessed by a residual metric, which computes the sum of squared distances between point correspondences post stitching (see *Figure 6a*). A median residual of <5 pixels was achieved for sections from all our datasets (top figure in *Figure 6b–e*), which is a requirement for successful 3D segmentation (*Macrina et al., 2021*) in addition to having no other stitching issues as described above. We aimed at 5 pixels (20 nm) as the target accuracy of the stitching because it is 10 times smaller than the average diameter of a spine neck (*Arellano et al., 2007*) and half the diameter of very thin spine necks. The violin plots in *Figure 6* depict the density distribution of the median residual values computed for every serial section from our datasets and are grouped by the acquisition systems. It can be seen that the density of distribution is below the threshold value (the horizontal line in these plots), indicating the stitching quality of the serial sections. A small number of sections reported high residuals even with the optimized set of solver parameters (*Figure 6b–e*). An attempt to re-montage them with parameters that will reduce the residuals resulted in distorting individual tile images. Hence, these sections were montaged using a set of parameters that produces a montage with less distorted tiles and a residual that can be tolerated by the 3D fine alignment process and further segmentation. Overall, we aim to achieve high-fidelity stitching by attempting to keep the residuals within the threshold, while preserving the image scales in both $x$ and $y$ closer to 1 (*Figure 6*) and occasionally allowing montages with residuals above the threshold.

The global 3D alignment process produces a volume that is 'roughly' aligned as the point correspondences are generated from montages materialized at 1% scale. This rough alignment provides a good initial approximation for fine alignment of the volume and for generating point correspondences at higher resolutions. The quality of global nonlinear 3D alignment is measured by computing

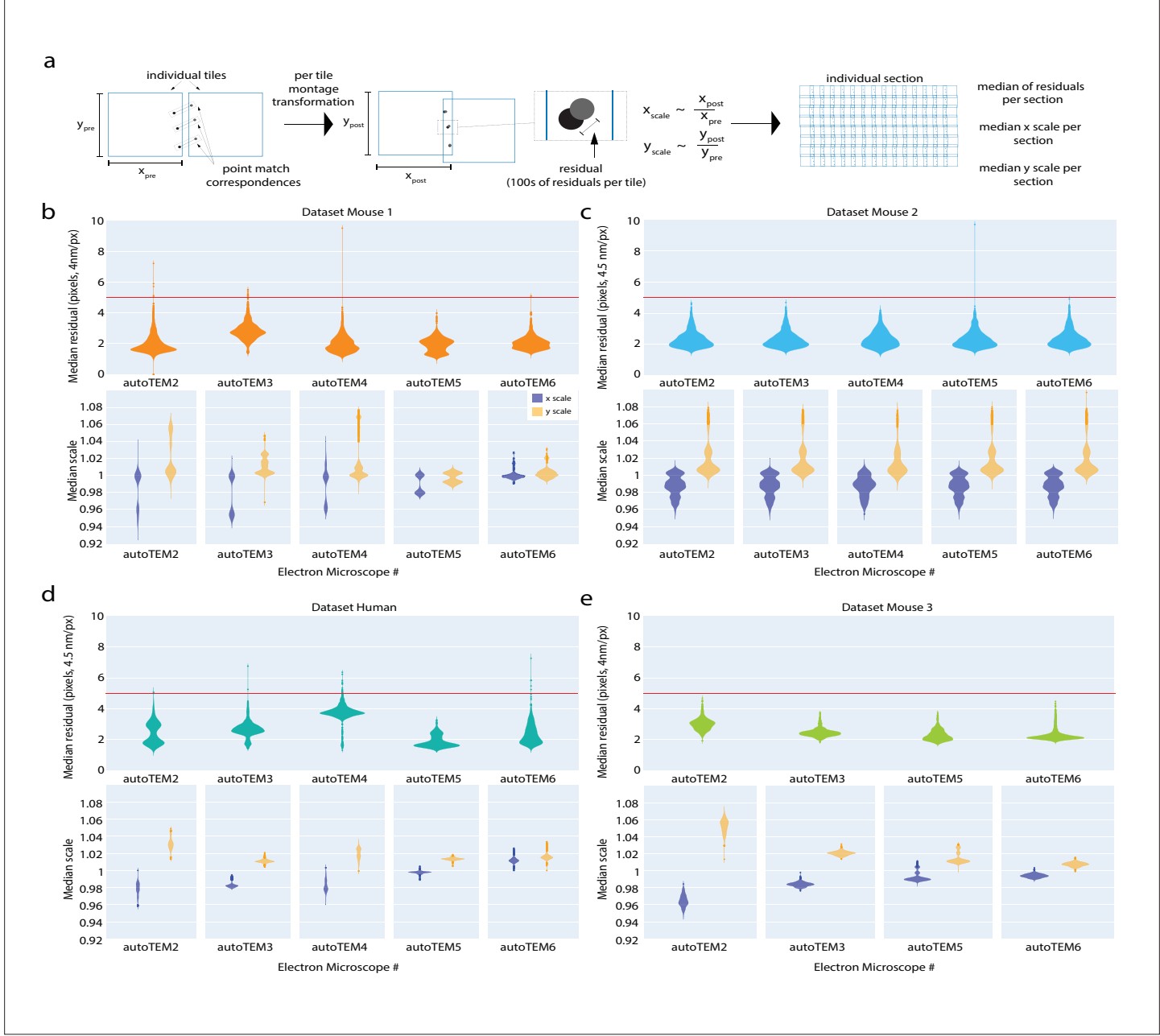

**Figure 6.** Performance of 2D stitching pipeline. (**a**) Schematic diagram explaining the computation of residuals between a pair of tile images post stitching. Residuals is a metric that is used to assess the quality of stitching in our pipeline. (**b–e**, top): panels (**b–e**) show the median of tile residuals per section grouped by their acquisition transmission electron microscopy (TEM). The horizontal line in these figures marks the threshold value that is set to assess the quality of stitching. *Table 2* shows the median residual values in *nm* for all our datasets. (**b–e**, bottom): panels (**b–e**) show the median $x$ and $y$ scale distribution of the tile images for all the datasets grouped by their acquisition system. The $x$ and $y$ scales of the tile images post 2D stitching indicate the level of deformation that a tile image undergoes post stitching – an indicator of the degree of quality of the 2D montaged section. (**b**) Mouse dataset 1. (**c**) Mouse dataset 2. (**d**) Human dataset. (**e**) Mouse dataset 3.

the angular residuals between pairs of sections (within a distance of three sections in *z*). The angular residual is computed using the point correspondences between a section and its neighbors. The angular residual is defined as the angle between two vectors formed by a point coordinate (from first section) and its corresponding point coordinate from a neighboring section. The origin of the two vectors is defined as the centroid of the first sections' point coordinates. The median of the angular residuals is reported as a quality metric for the global 3D alignment for our datasets (*Figure 7f*).

**Table 2.** Performance of 2D stitching pipeline.
Table shows the values of metrics computed to assess the quality of the stitched montages from all our datasets. The median residual value and the median scale in the x-axis and y-axis measures the accuracy of the alignment and the deformation factor of the individual tile images.

| Dataset | Median residuals (pixels) | Median residuals (nm) | Median scale (x) | Median scale (y) |
|---|---|---|---|---|
| Human dataset | 2.59 | 12.38 | 0.98 | 1.02 |
| Mouse dataset 1 | 2.10 | 8.36 | 0.99 | 1.00 |
| Mouse dataset 2 | 2.21 | 10.56 | 0.98 | 1.00 |
| Mouse dataset 3 | 2.37 | 9.27 | 0.99 | 1.01 |

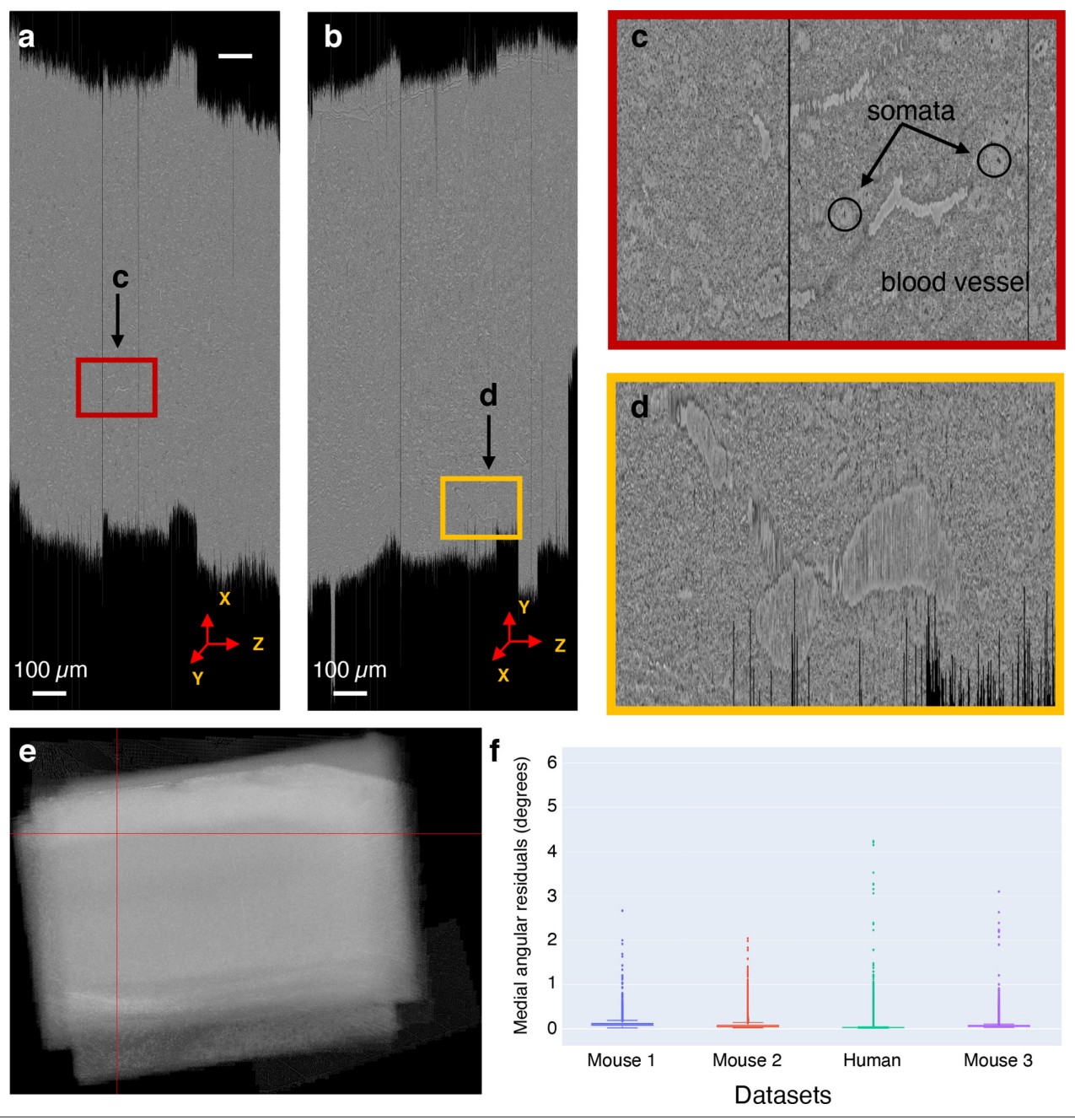

**Figure 7.** Global nonlinear 3D aligned volume of the mouse dataset 1. (**a**) View of the global nonlinear 3D aligned volume from the $xz$ plane. The figure shows the view of the global nonlinear 3D alignment of the sections with the volume sliced at position marked by the red lines in (**e**). (**b**) View of the global nonlinear 3D aligned volume from the $yz$ plane. Figure shows the view of the volume sliced at position marked by the red lines in (**e**). (**c**) Zoomed-in area from (**a**) showing the quality of global nonlinear 3D alignment in the $xz$ plane. (**d**) Zoomed-in area from (**b**) showing the quality of global nonlinear 3D alignment in the $yz$ plane. (**e**) Maximum pixel intensity projection of the global nonlinear 3D aligned sections in the z-axis showing the overall alignment of sections within the volume. The red lines represent the slicing location in both $xz$ and $yz$ plane for the cross-sectional slices shown in (**a**) and (**b**). (**f**) A plot showing the distribution of median angular residuals from serial sections grouped by the dataset.

The online version of this article includes the following figure supplement(s) for figure 7:

**Figure supplement 1.** Global nonlinear 3D aligned volume of the mouse dataset 2.

**Figure supplement 2.** Global nonlinear 3D aligned volume of the human dataset.

**Figure supplement 3.** Global nonlinear 3D aligned volume of the mouse dataset 3.

**Table 3.** Processing time comparison between acquisition system and Assembly Stitching and Alignment Pipeline (ASAP).

The acquisition times shown are based on serial sections imaged using five different serial section transmission electron microscopies (ssTEMs) running in parallel. The stitching time for all the datasets includes the time it took to stitch all the serial sections including semi-automated quality control (QC) and reprocessing sections that failed QC on the first run and the global 3D alignment. The stitching was done in a noncontinuous fashion that included correctly uploading/reuploading corrupted, duplicate sections, etc. Each section was stitched using a single node from the compute cluster. The different processing times of the different datasets reflect the optimization of the pipeline over time, while still keeping a throughput in pace with imaging acquisition.

| Dataset | No. of sections | Acquisition time (months) | ASAP processing time (includes manual processing times) |
|---|---|---|---|
| Mouse dataset 1 | 26,500 | 6 | 4 months |
| Mouse dataset 2 | 17,584 | 3 | 6 weeks |
| Human dataset | 9,661 | 2 | 4 weeks |
| Mouse dataset 3 | 17,309 | 3 | 10 days |

The quality metric ensures a high-quality global nonlinear 3D alignment of the sections in all three (*xy*, *yz*, *zx*) planes of the volume (see *Figure 7* for global nonlinearly 3D aligned slices from mouse dataset 1 and *Figure 7—figure supplements 1–3* for slices from other datasets). For the datasets described in this article, this global alignment was the initialization point for the fine alignment done outside ASAP with SEAMLeSS (*Macrina et al., 2021*). An illustration of the fine-aligned volume using SEAMLeSS on mouse dataset 1 can be found at https://www.microns-explorer.org/cortical-mm3. The infrastructure present in ASAP can be however extended to 'fine' alignments because ASAP is ready to implement 3D transformation both at the level of sections and at the level of individual image tiles. The quality of the fine alignment will depend on the transform that the user chooses to implement, ASAP is just a framework/vehicle for that transform.

*Table 3* provides a comparison of both dataset acquisition times and their volume assembly. The acquisition times represent the serial sections imaged using five different ssTEMs running in parallel. Each of the dataset processing times is under the same infrastructure settings (see 'BlueSky workflow engine for automated processing' for details on hardware setting), but with several optimizations implemented in ASAP with every dataset. The ASAP processing times also include the manual QC processing time duration. For each dataset, the manual QC processing time is roughly a few minutes per serial section, but has not been quantified for an accurate estimation that can be reported here. All of our datasets were processed in a time frame that matches or exceeds the acquisition time, thus achieving high-throughput volume assembly.

## Application to other imaging pipelines: Array tomography

The software infrastructure described in this article can also be applied to fluorescence and multimodal datasets such as array tomography (*Figure 8*). Array tomography presents some unique challenges for image processing because imaging can be performed in both light and electron microscopy. In addition, multiple channels can be imaged simultaneously and multiple rounds of imaging can be performed on the same physical sections with light microscopy (*Collman et al., 2015*). To properly integrate all these images, in addition to the image processing steps of 2D stitching and alignment that apply to EM, the multiple rounds of light microscopy of the same section must be registered to one another, and the higher resolution EM data must be co-registered with the light microscopy data. Finally, alignments based on one set of images must be applied to the other rounds and/or modalities of data. The Render services allow for image processing steps to define new transformations on the image tiles without making copies of the data, including transformations that dramatically alter the scale of the images, such as when registering between EM and light microscopy data. The Render and point-match services provide a flexible framework for corresponding positions between tiles to be annotated, allowing those correspondences to be used as constraints in calculating the appropriate transformations at each step of the pipeline. The result is a highly multimodal representation of the

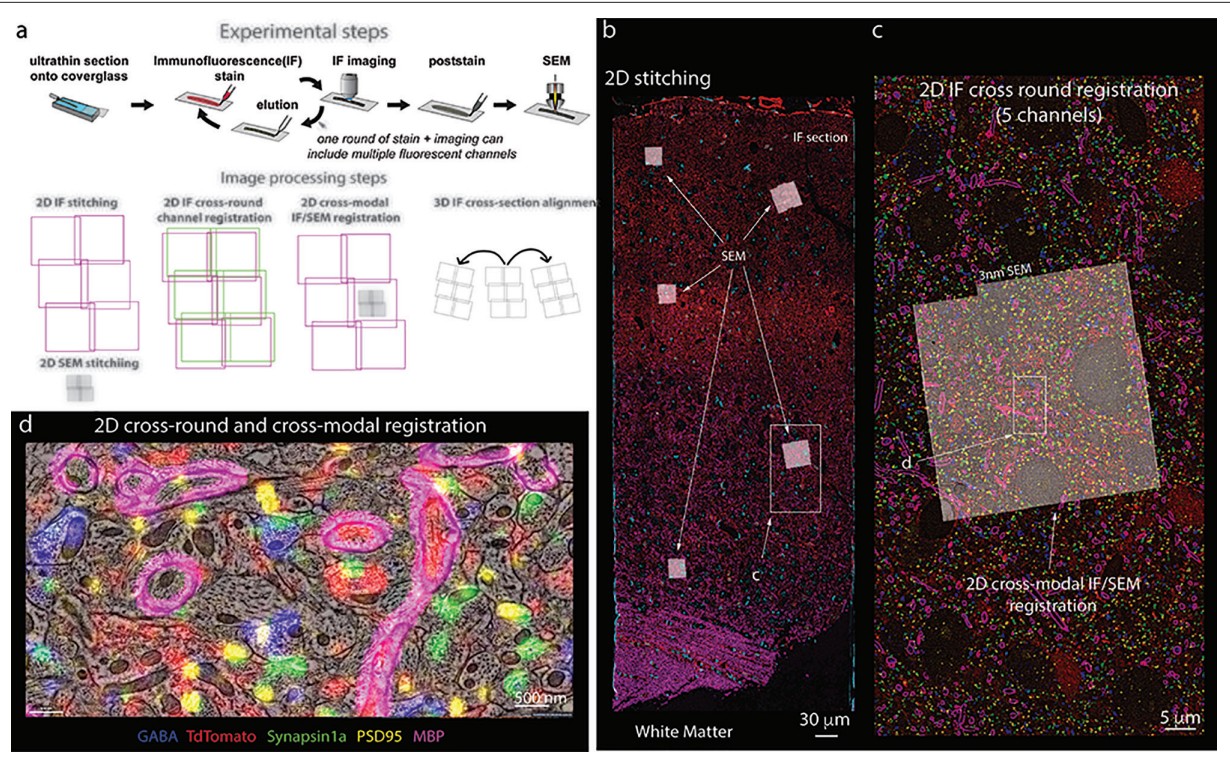

**Figure 8.** Stitching of multichannel conjugate array tomography data. (**a**, top) Experimental steps in conjugate array tomography: Serial sections are collected onto glass coverslips and exposed to multiple rounds of immunofluorescent (IF) staining, imaging, and elution, followed by post-staining and imaging under a field emission scanning electron microscopy (FESEM). (**a**, bottom) Schematic illustrating the substeps of image processing large-scale conjugate array tomography data. 2D stitching must be performed on each round of IF imaging and EM imaging. Multiple rounds of IF imaging of the same physical section must be registered together to form a highly multiplexed IF image of that section. The higher resolution by typically smaller spatial scale FESEM data must then be registered to the lower resolution but larger spatial scale IF data for each individual 2D section and FESEM montage. Finally, alignments of the data across sections must be calculated from the IF, or alternatively EM datasets. In all cases, the transformations of each of these substeps must be composed to form a final coherent multimodal, multiresolution representation of the dataset. (**b–d**) Screenshots of a processed dataset, rendered dynamically in Neuroglancer through the Render web services. (**b**) An overview image of a single section of conjugate array tomography data that shows the result of stitching and registering multiple rounds of IF an EM data. Channels shown are GABA (blue), TdTomato (Red), Synapsin1a (green), PSD95 (yellow), and MBP (purple). Small white box highlights the region shown in (**c**). (**c**) A zoom-in of one area of the section where FESEM data was acquired, small white box shows the detailed region shown in (**d**). (**d**) A high-resolution view of an area of FESEM data with IF data overlaid on top. One can observe the tight correspondence between the locations of IF signals and corresponding ultrastructural correlates, such a myelinated axons on MBP, and postsynaptic densities and PSD95.

dataset that can be dynamically visualized in multiple channels and resolutions at each step of the process through the integration of Render services with the Neuroglancer visualization tool (*Figure 8*).

## Discussion

The volume assembly pipeline ASAP was designed to produce high throughput if it is scalable, flexible, modular, upgradeable, and easily deployable in a variety of environments, including large-scale distributed systems. The pipeline leverages Render service's capability of processing by means of metadata operations and persisting data in databases. This largely facilitates multiple iterations of processing the data until a desired aligned volume is achieved. The need for rendering intermediate output is also eliminated at each iteration since the output can be dynamically rendered by applying the metadata associated with the images. This potentially saves computational time and resources in addition to increasing the throughput. Demonstrating its scalability, ASAP has been used to process several large-scale datasets, including a millimeter cube of mouse cortex that is already public at https://www.microns-explorer.org. Though ASAP is compatible with several strategies for fine alignment (*Figure 2*), the one used for all the datasets in this article was SEAMLeSS, which is described in *Macrina et al., 2021*.

The volume assembly pipeline maximizes the speed and quality of stitching and alignment on large-scale datasets. One of the main improvements is the addition of a parameter optimization module that generates optimized sets of parameters for 2D stitching. This parameter optimization was introduced for montaging in mouse dataset 2, mouse dataset 3, and the human dataset. The use of optimization parameters resulted in less distorted montages with residuals within acceptable threshold values. It also compensated for some deviation in lens distortion correction accuracy, while reducing the number of iterations of processing.

## Quality assessment

In a software pipeline that processes tens of millions of images, it is essential to have automated metrics of quality control. The statistical metrics such as MAD of the image scales to auto-detect deformed montages combined with detecting other stitching issues by the QC module facilitates faster processing while ensuring that the stitched sections meet the QC criteria. Also, early detection of poor point correspondences by the QC module drastically reduces the need for reprocessing montages through several iterations. About 2% of sections undergo this re-computation of point correspondences at a higher scale. Speed-up is also achieved by automating data transfer and ingestion into our volume assembly workflow from imaging. This is achieved by means of automatically querying the imaging database for sections that have been imaged and have passed imaging QC (W et al. 2020). The metadata of the QC passed sections are automatically ingested into the volume assembly workflow, which also triggers the stitching process. The imaging database was not developed during imaging of mouse dataset 1, hence the status of imaging and QC for each section was maintained in a spreadsheet and manually updated.

ASAP is capable of handling reimaged serial sections without overwriting the metadata for its earlier versions during processing. Also, the system is capable of handling missing sections (in case of serial section loss during sectioning or aperture burst/damage during imaging) and partial sections (sections that are cut partially from the volume). The missing sections are treated as 'gaps' in the volume and have minimal impact on the quality of alignment. Currently, the pipeline has successfully handled a gap of three consecutive sections (and five consecutive sections for the human dataset) in the volume. Feature-based computation of point correspondences is effective in finding features across sections with gaps between them and also robust to contrast and scale variations between image pairs.

The software stack includes capabilities to interface with different solvers through BigFeta including a scipy.sparse-based solver and the interfaces provided by PETSc (*Balay et al., 2019*, *Balay et al., 2021*, *Balay et al., 1997*). This has allowed us to nonlinearly globally 3D align an entire volume on a single workstation as well as on a distributed system. Our code base was also improved to allow for reprocessing individual sections that are reimaged and inserting them in existing global nonlinear 3D aligned volume. In addition to file storage, our software tools now support object stores using an S3 Application Program Interface (API) such as Ceph, Cloudian, and AWS, enabling real-time processing of large-scale datasets in the cloud as well as on-premises. The entire software stack is designed and developed using open-source dependencies and licensed under the permissive Allen Institute Software License. Also, our software stack and its modules are containerized allowing rapid deployment and portability. It also includes integration tests for each module for seamless development and code coverage. Automated processing of EM datasets can be accomplished with a custom workflow based on an open-source workflow manager (BlueSky) that is well suited to incorporate complex workflows with readable, flexibility workflow diagrams allowing rapid development.

## Image processing at the speed of imaging

The reconstruction of neural circuits requires high spatial resolution images provided by EM and drastic advances made in the field of EM connectomics (*MICrONS Consortium et al., 2021*; *Wetzel et al., 2016*; *MICrONS Consortium et al., 2021*; *Shapson-Coe et al., 2021*) that make it suitable for imaging large-scale EM volumes and producing dense reconstructions. ASAP aligns well with such large-scale EM volume production systems facilitating seamless processing of data through automated data ingestion, 2D stitching, 3D alignment, and QC – all chained together as a continuous process. Developing a pipeline that can produce 2D stitching at a rate better than imaging was the most challenging problem. In addition, we invested heavily to develop a set of software tools

that is modular, easily adaptable and upgradeable to new algorithms, computing systems and other domains, and able to run in a production-level setting.

The offline processing duration of all our datasets using ASAP has been shown to exceed the speed of imaging. ASAP is capable of processing the datasets in parallel with the imaging session with sufficient computational resources. The mouse dataset 1 was processed in parallel with imaging (stitching of serial sections) followed by a chunk-based global 3D alignment (first iteration). Efficient data transfer from the multiscope infrastructure, coupled with automated processing capabilities of ASAP, assisted in the processing of the mouse dataset 1 in parallel to imaging and at speeds that match the imaging. The *em stitch* software package leverages the GPU-based computations on the scope for imaging QC to stitch the montages on-scope. This accelerates the stitching process and the rapid feedback loop between imaging and volume assembly. Though our current processing rate already outperforms image acquisition, the next step is to perform the image processing in real time, ideally close to the microscopes and as images are collected. Such strategy has been proposed by Jeff Lichtman and colleagues (*Lichtman et al., 2014*), and there are many aspects of the work presented here that will facilitate transition to on-scope real-time stitching and alignment.

## Scaling to larger volumes and across image modalities

Our pipeline was developed with a focus on standardization and was built entirely with open-source libraries as an open-source package. Our intention is for others to use and potentially scale it beyond the work described in this article. As we demonstrate in *Figure 8*, the use of ASAP goes well beyond electron microscopy, and it is being used in fluorescent data as well. The modularity of ASAP can be leveraged to include GPU-based algorithms at various stages of the pipeline, thus paving the way for further increase in throughput. Processing in parallel with imaging, we were able to stitch and globally nonlinearly 3D align 2 PB of EM images from the 1 mm$^3$ mouse visual cortex at synapse resolution within a period of ~4 months, and other petascale datasets with a montaging rate exceeding the imaging rate. With improvements made to the pipeline, stitching and global nonlinear 3D alignment of a dataset similar in size took just 10 days of processing time for mouse dataset 3. This throughput makes the volume assembly pipeline suitable for processing exascale datasets that spans larger cortical areas of the brain across species. Although the pipeline was designed for EM connectomics, it can be easily adapted to process datasets from any other domain of image sets that share the basic underlying assumptions in imaging.

## Methods
### Imaging with electron microscopy

Three of the samples processed by the infrastructure described in this article originated from mice. All procedures were carried out in accordance with the Institutional Animal Care and Use Committee approval at the Allen Institute for Brain Science with protocol numbers 1503, 1801, and 1808. All mice were housed in individually ventilated cages, 20–26°C, 30–70% relative humidity, with a 12 hr light/dark cycle. Mouse genotypes used were as follows: mouse 1, Slc-Cre/GCaMP6s (JAX stock 023527 and 031562); mouse 2, Slc17a7-IRES2-Cre/CamK2a-tTA/Ai94 (JAX stock 023527, 024115); mouse 3, Dlx5-CreER/Slc-Cre/GCaMP6s (JAX stock 010705, 023527, 031562).

Preparation of samples was performed as described earlier (W et al. 2020); briefly, mice were transcardially perfused with a fixative mixture of 2.5% paraformaldehyde and 1.25% glutaraldehyde in buffer. After dissection, slices were cut with a vibratome and post-fixed for 12–48 hr. Human surgical specimen was obtained from a local hospital in collaboration with local neurosurgeon. The sample collection was approved by the Western Institutional Review Board (protocol # SNI 0405). The patient provided informed consent, and experimental procedures were approved by the hospital institute review boards before commencing the study. A block of tissue ~ 1 × 1 × 1 cm of anteromedial temporal lobe was obtained from a patient undergoing acute surgical treatment for epilepsy. This sample was excised in the process of accessing the underlying epileptic focus. Immediately after excision the sample was placed into a fixative solution of 2.5% paraformaldehyde, 1.25% glutaraldehyde, 2 mM calcium chloride, in 0.08 M sodium cacodylate buffer for 72 hr. The samples were then trimmed and sectioned with a vibratome to 1000-μm-thick slices and placed back in fixative for ~96 hr. After fixation, slices of mouse and human were extensively washed and prepared for reduced osmium

treatment (rOTO) based on the protocol of *Hua et al., 2015*. Potassium ferricyanide was used to reduce osmium tetroxide and thiocarbohydrazide (TCH) for further intensification of the staining. Uranyl acetate and lead aspartate were used to enhance contrast. After resin embedding, ultrathin sections (40 nm or 45 nm) were manually cut in a Leica UC7 ultra-microtome and an RMC Atum-tome. After sectioning, the samples were loaded into the automated transmission electron micro-scopes (autoTEM), and we followed the TEM operation routine (described in *Wetzel et al., 2016* and *MICrONS Consortium et al., 2021*) to bring up the HT voltage and filament current and then align the beam. Calibration of the autoTEM involved tape and tension calibration for barcode reading, measuring beam rotation and camera pixels, and stage alignment. Then, EM imaging was started. The mouse datasets were obtained from primary visual cortex and higher visual areas, and the human dataset was obtained from the Medial Temporal Gyrus (MTG).

## Image catcher (Aloha) service

Aloha is a core component of our acquisition infrastructure designed to facilitate the transfer and preprocessing of images intended for the image processing workflow. Aloha is implemented as a scale-out Python web service using flask/gunicorn. This service is designed to accept image arrays defined by a flat-buffers protocol and atomically write them in a designated location in losslessly compressed tiff format. While the array is in memory, the service also writes progressively downsam-pled versions of that image (MIPmaps) to another designated location. By using the uri-handler library (*Torres, 2021a*), Aloha can write to various cloud providers and on-premises object storage systems as well as file system-based storage. The Aloha library includes a set of client scripts that allow uploading from an existing autoTEM-defined directory as well as utilities to encode numpy arrays for the REST API. Aloha web service is configured to interact with the piTEAM's TEMdb backend and tracks the state of transfers in a set of custom fields. In the automated workflow, a process queries these fields in order to ingest completed montage sets to the volume assembly workflow. Aloha can be easily replaced with a data transfer module of choice based on the imaging infrastructure and the volume assembly workflow allowing for modularity.

## Render services

The Render services are a core component of the infrastructure. They provide the main logic for image transformation, interpolation, and rendering. They also provide a rich API:

- A REST API for creating and manipulating collections of tiles or image 'boxes' (also called canvases; canvases are regions that can span multiple and partial tiles).
- A REST API for accessing image tile, section, and stack meta information; for example, the number of tiles, dimensions, ids, and camera setup.
- A REST API and core logic for rendering/materializing image tiles/canvases, arbitrary regions that span a number of (or partial) tiles, or even whole sections. In that capacity, it is used to reflect the current state of any given tile collection. (This is invaluable to proofreading interme-diate stitching results.) In combination with dynamic rendering (i.e., rendering that is not based on materializing image files to storage), the Render services support lightweight web pages with feedback to detect imaging and stitching issues.

The Render services are backed by a MongoDB document store that contains all tile/canvas data including tile transformations. Both the Render services and the MongoDB document store are supported by dedicated hardware. The Render services code base is available and documented at https://github.com/saalfeldlab/render; *Preibisch, 2022*.

## Point-match service

A time-consuming and CPU-intensive process in the volume assembly pipeline is the computation of point correspondences between image tile pairs since this is the only stage of processing where the image data is read in memory besides the process of rendering the aligned volume to disk. Persisting this data is therefore invaluable. Robust rotation and contrast invariant correspondence candidates are generated using SIFT (*Lowe, 2004*). These candidates are then filtered by their consensus with respect to an optimal geometric transformation, in our case an affine transformation. We use a local optimization variant of RANSAC *Fischler and Bolles, 1981* followed by robust regression *Saalfeld et al., 2010*. Local optimization means that, instead of picking the 'winner' from a minimal set of

candidates as in classic RANSAC, we iteratively optimize the transformation using all inlier candidates and then update the inlier set. The 'winner' of this procedure (the largest set of inliers) is then further trimmed by iteratively removing candidates with a residual larger than 3 standard deviations of the residual distribution with respect to the optimal transformation and then reoptimizing the transformation. We use direct least-squares fits to optimize transformations. The computed point correspondences are stored in a database and can be retrieved/modified using the point-match service. The advantage of such a database is that it is agnostic to the source of point correspondences. Therefore, it can receive input from the point-match generator, regardless of the method of point-match generation such as SURF, SIFT, phase correlation, etc.

### Render-python API

The other core component of the software stack includes *render-python*, a Python API client and transformation library that interacts with both asap-modules and the Render services. The render-python components interact with Render service Java clients that perform computationally expensive operations locally to avoid taxing Render services running on centralized shared hardware.

*Render-python* is a python-based API client and transformation library that replicates the data models in the Render services. While Render services utilize the mpicbg Fiji library to implement transformations, render-python reproduces these using using numpy to enable analysis in a Python ecosystem. Render-python is continuously integration tested against Render for compatibility and provides dynamic access to the database and client scripts provided by Render. The source code for *render-python* is available at https://github.com/AllenInstitute/render-python (*Collman et al., 2022*).

Besides render-python, ASAP interfaces with other tools for solving transformations and visualizations. A description of these tools is as follows:

> BigFeta: The *BigFeta* package implements a Python-based sparse solver implementation of alignment problems based on the formulation in EMAligner (*Khairy et al., 2018*). In addition to optimizations and new transform functionality, BigFeta extends the previous approach to use PETSc (petsc.org) for scalable computation and allows input and output using render-python objects, as well as JSON file, MongoDB document store, and Render services interaction.
> em stitch: *em stitch* includes tools based on BigFeta and render-python for use as a standalone montage processing package without connecting to a database or REST API, ideal for online processing utilizing the same hardware running a microscope. Importantly, *em stitch* includes a module to derive a mesh-based lens distortion correction from highly overlapping calibration acquisitions and pre-generated point correspondences.
> vizrelay: *vizrelay* is a configurable microservice designed to build links from a Render services instance to a Neuroglancer-based service and issue redirects to facilitate visualization. This provides a useful mechanism for setting Neuroglancer defaults, such as the extent of the volume or color channel options when reviewing alignments.

### ASAP modules

The ASAP volume assembly pipeline includes a series of modules developed using Python and the render-python library that implement workflow tasks with standardized input and output formatting. The source code for ASAP modules is available at https://github.com/AllenInstitute/asap-modules; *Mahalingam, 2022*.

The submodules in ASAP include scripts to execute a series of tasks at each stage of the volume assembly pipeline. Some of the workflow tasks included in ASAP are as follows:

- *asap.dataimport*: Import image (tile) metadata to the Render services from custom microscope files, generate MIPmaps and update the metadata, render downsampled version of the montaged serial section.
- *asap.mesh_lens_correction*: Include scripts to compute the lens distortion correction transformation.
- *asap.pointmatch*: Generate tile pairs (see *Figure 2d*) and point correspondences for stitching and alignment.
- *asap.point_match_optimization*: Find the best possible set of parameters for a given set of tile pairs.
- *asap.solver*: Interface with BigFeta solver for stitching the serial sections.

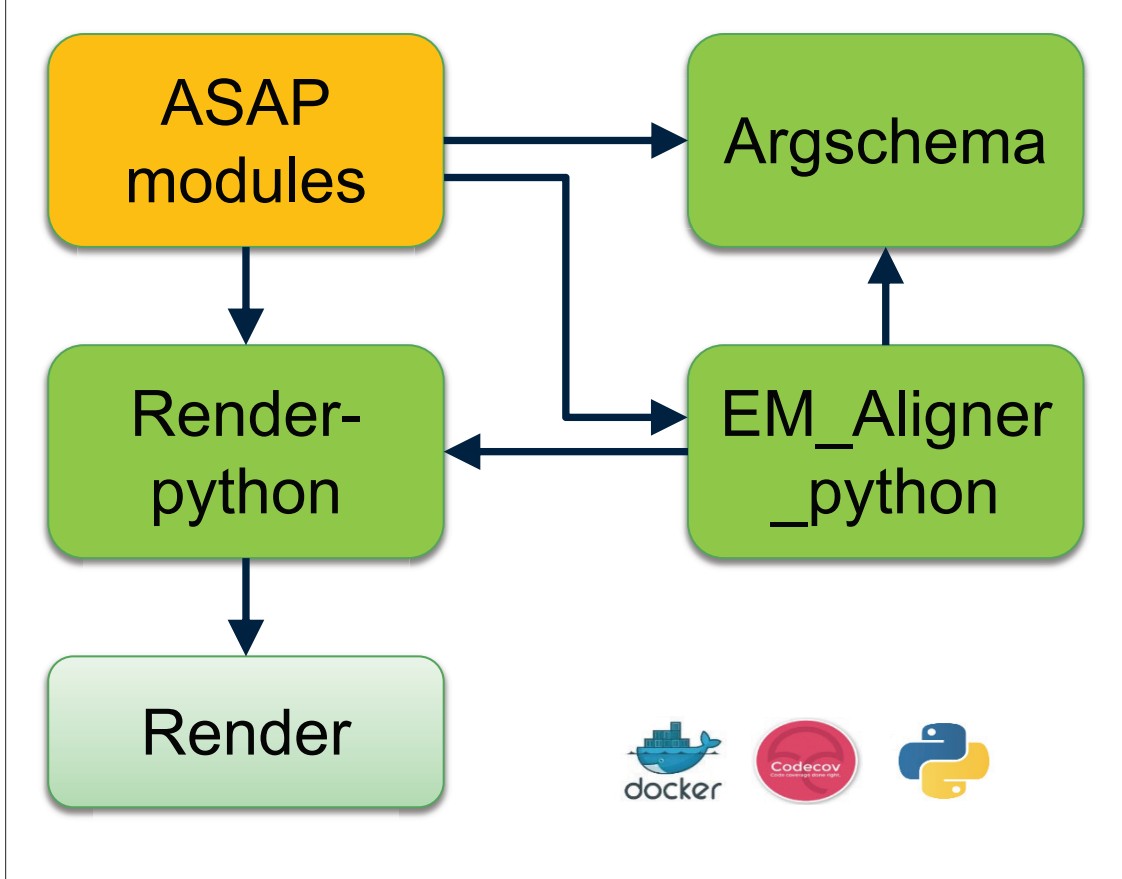

**Figure 9.** Set of software tools developed to perform petascale real-time stitching.

- *asap.em_montage_qc*: Generate QC statistics on the stitched sections as explained in 'Automated petascale stitching.'
- *asap.rough_align*: Compute per-section transformation for 3D alignment and scale them to match their original layered montage collection and generate new metadata describing the alignment at full resolution.
- *asap.register*: Register an individual section with another section in a chunk. This module is typically used to align reimaged sections to an already aligned volume.
- *asap.materialize*: Materialize final volume as well as downsampled versions of sections in a variety of supported formats.

ASAP modules are schematized for ease of use with *argschema*, an extension of the marshmallow Python package that allows marshaling of command-line arguments and input files. ASAP modules interact with other tools that comprise the peta-scale stitching and alignment software tools ecosystem (see *Figure 9*).

## Montage parameter optimization

In two dimensions (*x, y*), BigFeta implements the optimization described by *Khairy et al., 2018* as the following regularized least-squares problem:

$$\mathbf{K}\mathbf{t}_x = \mathbf{L}_x$$
$$\mathbf{K}\mathbf{t}_y = \mathbf{L}_y \tag{1}$$

with

$$\mathbf{K} = \mathbf{A}^T\mathbf{W}\mathbf{A} + \lambda$$
$$\mathbf{L}_{(x,y)} = \lambda\mathbf{t}_{(x0,y0)} + \mathbf{A}^T\mathbf{W}\mathbf{b}_{(x,y)} \tag{2}$$

where $\mathbf{t}_{(x,y)}$ are the unknowns for which to solve – these are interpreted to define the parameters of tile transformations, $\mathbf{A}$ is an $m \times n$ matrix from $m$ point correspondences derived from $2n$ total unknowns, $\mathbf{W}$ is an $m \times m$ diagonal matrix weighting the point correspondences, $\lambda$ is an $n \times n$ diagonal matrix containing regularization factors for the unknowns, $\mathbf{t}_{(x0,y0)}$ is the initialization for the unknowns against which the regularization penalizes, and $\mathbf{b}_{(x,y)}$ is a right-hand-side term to the unknowns introduced to generalize the method to additional transformation models.

Bigfeta allows the regularization parameter $\lambda$ to differently constrain distinct terms of a given transformation such as the translation, affine, and polynomial factors on an individual tile basis.

Montage quality in ASAP is evaluated by metrics of residuals and rigidity of the output montage (**Figure 4**, **Figure 5**). For tile deformations that are well-defined by an affine transformation, these metrics are most impacted by the translation and affine regularization parameters ($\lambda$) used in the BigFeta solver step (**equation 2**). As the optimal configuration of these values can be impacted by the accuracy of the initialization as well as the weight and distribution of point correspondences, it is sometimes necessary to re-evaluate the regularization parameters for different imaging, tissue, or preprocessing conditions. We provide an 'optimization' module, asap.solver.montage_optimization, which leverages the fast solving capabilities of BigFeta to sweep across a range of regularization parameters and provide an acceptable set of parameters given targets for correspondence residual in pixels and tile scale MAD value.

In each dataset where montage optimization was used, we found that a MAD cutoff of 0.005 in both $x$ and $y$ was usually sufficient to provide a group of acceptable montages. However, in some cases individual montages or sets of montages must be run with relaxed criteria – for these cases, candidate montages were usually found by increasing the MAD statistic to 0.006 and 0.007 in *x* and *y*, respectively. In our experience, these sets of montages seem to share a lens distortion model, and we assume that this inadequate model requires additional deformation at the tile and montage level. To help with these cases, our implementation of montage optimization has the option to iteratively relax these constraints to a predetermined boundary such that inadequate cutoffs can be increased until there are a desired number of candidate montages.

## 3D realignment

A common use case after 3D global alignment involves realigning a subset of the dataset while maintaining the global alignment reached for the rest of the volume. The asap.solver.realign_zs module implements this operation by increasing the $\lambda$ parameters in **equation 1** for neighboring sections while allowing custom inputs for the sections that need to be realigned. As such, it is possible to integrate re-montaged sections, re-computed point correspondences, or more deformable transformations into an existing 3D alignment without requiring changes on the global scale. For all the datasets presented in this article, after global alignment the data was then transferred for fine alignment using SEAMLeSS (**Macrina et al., 2021**). The fine alignment was performed by the team of Sebastian Seung in Princeton or ZettaAI.

## Chunk fusion

The asap.fusion package provides modules to support chunk-based 3D alignment workflows. The 3D aligned chunks can then be fused together. asap.fusion.register_adjacent_stack provides utilities to register overlapping 3D aligned chunks using translation, rigid, similarity, or affine transformations. Then, given a JSON-defined tree describing the layout and relative transformation between chunks, asap.fusion.fuse_stacks will assemble metadata representing a combined volume using Render's 'InterpolatedTransform' to interpolate between independently optimized transformations in the overlap region of two chunks.

## Materialization

Alignment through BigFeta produces tile metadata that can be interpreted by Render and its clients to produce 'materialized' representations of the stack. These representations are rendered client-side, having transformations applied and flattening overlapping tiles. The output of materialization can be in a number of file formats including n5 and the CATMAID large data tilesource (**Schneider-Mizell et al., 2016**; **Saalfeld et al., 2009**), which are included in Render. It is also possible to implement custom output formats based on available Python libraries using render-python to access Render's

client scripts. As SEAMLeSS expects data in the cloud-volume-compatible Neuroglancer precomputed format Neuroglancer, the datasets in *Table 1* were materialized with a Python script that uses render-python to write to Neuroglancer precomputed format using the cloud-volume library.

## BlueSky workflow engine for automated processing

The automated workflow engine called BlueSky was developed in Django backed by a PostgreSQL database with stable backups, graceful restarts, and easy migrations. It provides a web-based user interface for the user to visualize, run, and edit running jobs at various stages in the workflow. BlueSky uses Celery and RabbitMQ to run workflow tasks in diverse computing environments, from local execution on a workstation to remote execution using a compute cluster (PBS, MOAB, SLURM). BlueSky is flexible in terms of designing complex workflows as the workflow diagrams (see *Appendix 1—figure 1*, *Appendix 2—figure 1*) can be specified in readable formats such as YAML, JSON, or Django allowing rapid development. BlueSky can be used for many different purposes, but for the image processing task related to this article the workflow includes the following steps: (1) ingest montage sets, (2) generate MIPmaps, (3) apply MIPmaps, (4) wait for the assigned lens correction transform, (5) apply the lens correction transform, (6) extract tile pairs for determining point correspondences, (7) generate 2D montage point correspondences, (8) run the 2D montage solver, (9) automatically check for defects, (10) place potential defects in a manual QC queue, and (11) generate downsampled montage. BlueSky is publicly available on GitHub (https://github.com/AllenInstitute/blue_sky_workflow_engine). The volume assembly workflow is designed to use BlueSky workflow engine for processing our datasets. The custom EM volume assembly workflow (https://github.com/AllenInstitute/em_imaging_workflow; *Torres et al., 2021b*) facilitates continuous processing of the datasets at speeds that match or exceed data acquisition rates (see *Appendix 3—figure 1*, *Appendix 4—figure 1*).

For all our datasets, BlueSky utilized three different kinds of hardware nodes in our HPC cluster. The configurations are as follows:

- 2× Intel Xeon 2620 processor with 256 GB RAM
- 2× Intel Xeon 2630 processor with 256 GB RAM
- 2× Intel Xeon Gold 6238 processor with 512 GB RAM

All our datasets were processed using a maximum of 50 nodes to perform 2D stitching of sections in parallel. A combination of the nodes with the above configuration was used in processing.

## Array tomography alignment

To register the light and electron array tomography data into a registered conjugate image stack, we developed a set of modules that used a manual process for image registration. Briefly, one module (make_EM_LM_registration_projects_multi) created a TrakEm2 project in which the EM data is in one z-layer, and each light microscopy channel is in a different z-layer. From there, we created a blended view of DAPI and MBP stain to create an image with recognizable features between the two datasets. Users manually identified sets of correspondences between the images, including the centers of myelinated processes, the centers of mitochondria visible often in the autofluorescence of the DAPI channel, and spatially distinct regions of heterochromatin that appears bright in the DAPI channel and dark in the EM. Between 12 and 20 corresponding points were identified in each section, and TrackEm2 was used to fit as similarity transform to bring the images into register. A second module (import_LM_subset_from_EM_registration_multi) then exported the transformations saved in this Trakem2 project back into the render framework. We implemented this custom workflow outside of the main automated EM image processing pipeline so it is available in a separate repository (https://www.github.com/AllenInstitute/render-python-apps; *Collman, 2018*) within the submodule 'renderapps/cross_modal_registration'.

## Acknowledgements

We thank our project manager Shelby Suckow for her exceptional work in keeping us aligned and on time. We thank Allan Jones and Gerry Rubin for starting the conversation that led to this collaboration and for their support and leadership. We thank Hongkui Zeng and Christof Koch for their support and leadership. We thank D Brittain, M Scott, and J Borseth for their help in collecting and imaging the

EM datasets. We thank Sebastian Seung, Thomas Macrina, Nico Kemnitz, Manuel Castro, Dodam Ih, and Sergiy Popovych from Princeton University and ZettaAI for discussions and feedback on image processing strategies and improvements. We thank Brian Youngstrom, Stuart Kendrick, and the Allen Institute IT team for support with infrastructure, data management, and data transfer. We thank Jay Borseth, DeepZoom LLC, for his contributions to *em stitch*. We thank Andreas Tolias, Jacob Reimer, and their teams at the Baylor College of Medicine for providing mice used for electron microscopy. We thank Saskia de Vries, Jerome Lecoq, Jack Waters, and their teams at the Allen Institute for providing mice used for electron microscopy. This work was supported by the Intelligence Advanced Research Projects Activity (IARPA) of the Department of Interior/Interior Business Center (DoI/IBC) through contract number D16PC00004 and by the Allen Institute for Brain Science. The views and conclusions contained herein are those of the authors and should not be interpreted as representing the official policies or endorsements, either expressed or implied, of the funding sources including IARPA, DoI/IBC, or the US government. We wish to thank the Allen Institute founder, Paul G Allen, for his vision, encouragement, and support.

## Additional information

### Competing interests

Eric Perlman: has a competing interest in Yikes LLC. The other authors declare that no competing interests exist.

### Funding

| Funder | Grant reference number | Author |
| --- | --- | --- |
| Intelligence Advanced Research Projects Activity (IARPA) of the Department of Interior/Interior Business Center | D16PC00004 | Gayathri Mahalingam<br>Russel Torres<br>Daniel Kapner<br>Tim Fliss<br>Shamishtaa Seshamani<br>Rob Young<br>Samuel Kinn<br>JoAnn Buchanan<br>Marc M Takeno<br>Wenjing Yin<br>Daniel J Bumbarger<br>R Clay Reid<br>Forrest Collman<br>Nuno Macarico da Costa |

The funders had no role in study design, data collection and interpretation, or the decision to submit the work for publication.

### Author contributions

Gayathri Mahalingam, Software, Formal analysis, Validation, Investigation, Visualization, Methodology, Writing - original draft, Optimized the pipeline software (asap-modules) and executed the stitched and aligning of the datasets. Developed the software packages (asap-modules, render-python) and maintained the infrastructure and continuous integration testing used by the Render backed pipeline; Russel Torres, Data curation, Software, Formal analysis, Validation, Investigation, Visualization, Methodology, Writing - original draft, Optimized the pipeline software (asap-modules) and executed the stitched and aligning of the datasets. Primary developer of Aloha . Developed the software packages (asap-modules, render-python) and maintained the infrastructure and continuous integration testing used by the Render backed pipeline; Daniel Kapner, Software, Methodology, Writing – review and editing, Developed EM aligner; Eric T Trautman, Software, Methodology, Writing – review and editing, Primary developers of the Render services; Tim Fliss, Software, Methodology, Writing – review and editing, Developed BlueSky workflow manager and the volume assembly workflow; Shamishtaa Seshamani, Software, Methodology, Writing – review and editing, Developed the software packages (asap-modules, render-python) and maintained the infrastructure and continuous integration testing used by the Render backed pipeline. Generated Array tomography data; Eric Perlman, Software,

Methodology, Writing – review and editing, Developed multi-channel support for asap-modules and vizrelay. Developed the software packages (asap-modules, render-python) and maintained the infrastructure and continuous integration testing used by the Render backed pipeline; Rob Young, Project administration, Writing – review and editing; Samuel Kinn, Software, Writing – review and editing, Developed Aloha; JoAnn Buchanan, Methodology, Writing – review and editing, Contributed to the generation of mouse EM data. Contributed to the generation of human EM data; Marc M Takeno, Methodology, Writing – review and editing, Contributed to the generation of mouse EM data. Contributed to the generation of human EM data; Wenjing Yin, Methodology, Writing – review and editing, Contributed to the generation of mouse EM data. Contributed to the generation of human EM data; Daniel J Bumbarger, Methodology, Writing – review and editing; Ryder P Gwinn, Methodology, Writing – review and editing, Contributed to the generation of human EM data; Julie Nyhus, Methodology, Writing – review and editing, Contributed to the generation of human EM data; Ed Lein, Methodology, Writing – review and editing, Contributed to the generation of human EM data; Steven J Smith, Supervision, Funding acquisition, Methodology, Writing – review and editing; R Clay Reid, Conceptualization, Funding acquisition, Writing – review and editing; Khaled A Khairy, Conceptualization, Software, Supervision, Writing – review and editing, Primary developer of the MATLAB version of EM aligner. Conceptualized the stitching pipeline using Render services; Stephan Saalfeld, Conceptualization, Software, Supervision, Investigation, Writing – review and editing, Primary developers of the Render services; Forrest Collman, Software, Supervision, Writing – review and editing, Developed the software packages (asap-modules, render-python) and maintained the infrastructure and continuous integration testing used by the Render backed pipeline. Generated Array tomography data; Nuno Macarico da Costa, Conceptualization, Supervision, Funding acquisition, Investigation, Methodology, Writing - original draft, Project administration, Contributed to the generation of mouse EM data. Contributed to the generation of human EM data

### Author ORCIDs

Gayathri Mahalingam http://orcid.org/0000-0003-3718-5013
Russel Torres http://orcid.org/0000-0002-2876-4382
Eric T Trautman http://orcid.org/0000-0001-8588-0569
Eric Perlman http://orcid.org/0000-0001-5542-1302
Marc M Takeno http://orcid.org/0000-0002-8384-7500
Steven J Smith http://orcid.org/0000-0002-2290-8701
R Clay Reid http://orcid.org/0000-0002-8697-6797
Khaled A Khairy http://orcid.org/0000-0002-9274-5928
Stephan Saalfeld http://orcid.org/0000-0002-4106-1761
Forrest Collman http://orcid.org/0000-0002-0280-7022
Nuno Macarico da Costa http://orcid.org/0000-0003-2001-4568

### Ethics

Human subjects: Human surgical specimen was obtained from local hospital in collaboration with local neurosurgeon. The sample collection was approved by the Western Institutional Review Board (Protocol # SNI 0405). Patient provided informed consent and experimental procedures were approved by hospital institute review boards before commencing the study.

All procedures were carried out in accordance with Institutional Animal Care and Use Committee approval at the Allen Institute for Brain Science with protocol numbers 1503, 1801 and 1808.

### Decision letter and Author response

Decision letter https://doi.org/10.7554/eLife.76534.sa1
Author response https://doi.org/10.7554/eLife.76534.sa2

## Additional files

### Supplementary files

• Transparent reporting form

### Data availability

The current manuscript describes is a software infrastructure resource that is being made publicly available. The manuscript is not a data generation manuscript. Nevertheless, one of the datasets

used is already publicly available on https://www.microns-explorer.org/cortical-mm3#em-imagery with available imagery and segmentation (https://tinyurl.com/cortical-mm3). Moreover cloud-volume (https://github.com/seung-lab/cloud-volume) can be used to programmatically download EM imagery from either Amazon or Google with the cloud paths described below. The imagery was reconstructed in two portions, referred to internally by their nicknames 'minnie65' and 'minnie35' reflecting their relative portions of the total data. The two portions are aligned across an interruption in sectioning. minnie65: AWS Bucket: precomputed: https://bossdb-open-data.s3.amazonaws.com/iarpa_microns/minnie/minnie65/em Google Bucket: precomputed: https://storage.googleapis.com/iarpa_microns/minnie/minnie65/emminnie35: AWS Bucket: precomputed: https://bossdb-open-data.s3.amazonaws.com/iarpa_microns/minnie/minnie35/em Google Bucket: precomputed: https://storage.googleapis.com/iarpa_microns/minnie/minnie35/em. We have also made available in Dryad raw data of the remaining datasets https://doi.org/10.5061/dryad.qjq2bvqhr.

The following dataset was generated:

| Author(s) | Year | Dataset title | Dataset URL | Database and Identifier |
|---|---|---|---|---|
| Macarico da Costa N, Mahalingam G, Torres R, Buchanan J, Takeno M, Yin W, Bumbarger D, Collman F, Reid R | 2022 | ASAP-TEM-sample | https://doi.org/10.5061/dryad.qjq2bvqhr | Dryad Digital Repository, 10.5061/dryad.qjq2bvqhr |

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

# Appendix 1

```
graph:
    - [ "ingest_tile_sets", [ "generate_lens_correction", "generate_em_montage" ] ]
    - [ "generate_lens_correction", [ "move_reference_set" ] ]
    - [ "generate_em_montage", [ "generate_mipmaps" ] ]
    - [ "generate_mipmaps", [ "apply_mipmaps" ] ]
    - [ "apply_mipmaps", [ "wait_for_lens_correction" ] ]
    - [ "move_reference_set", [ "wait_for_lens_correction" ] ]
    - [ "wait_for_lens_correction", [ "apply_lens_correction_new" ] ]
    - [ "apply_lens_correction_new", [ "create_tile_pairs" ] ]
    - [ "create_tile_pairs", [ "2d_montage_point_match" ] ]
    - [ "2d_montage_point_match", [ "solver" ] ]
    - [ "solver", [ "detect_defects" ] ]
    - [ "detect_defects", [ "wait_for_manual_qc" ] ]
    - [ "wait_for_manual_qc", [ "downsample", "move_raw_montage_set" ] ]
    - [ "downsample", [ "alignment" ] ]
```

**Appendix 1—figure 1.** Intelligence Advanced Research Projects Activity (IARPA) MICrONS phase 2 montage workflow with support for lens correction and manual intervention.

## Appendix 2

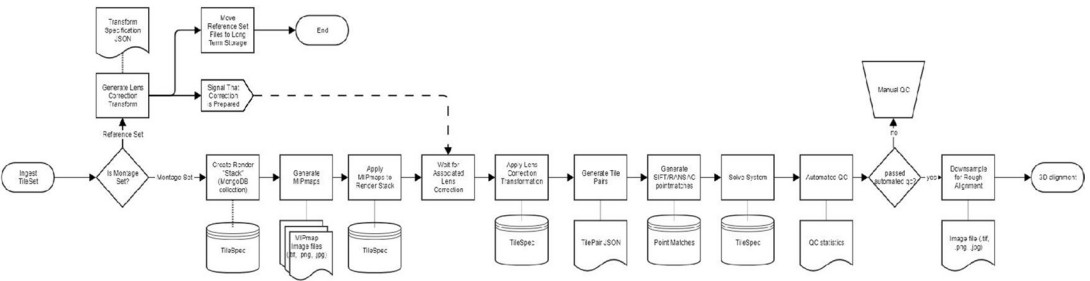

**Appendix 2—figure 1.** Electron microscopy (EM) workflow diagram from 10 represented in YAML as DAG.

## Appendix 3

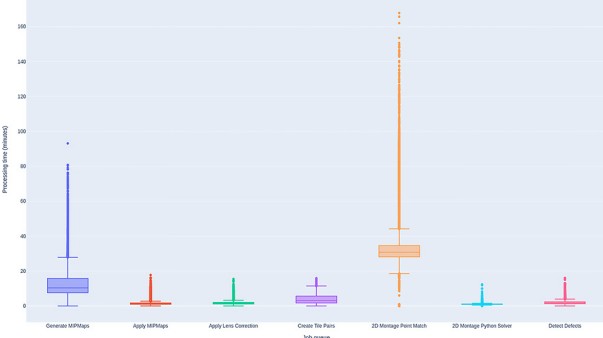

**Appendix 3—figure 1.** Performance of BlueSky workflow manager. Time spent by the sections from all the datasets in each job queue in the montaging workflow. The processing times include the duration between the time the job started running in a node and the time the node releases the job as successfully completed. Processing times shown are based on running the job in a single computing node in every job queue.

**Appendix 4**

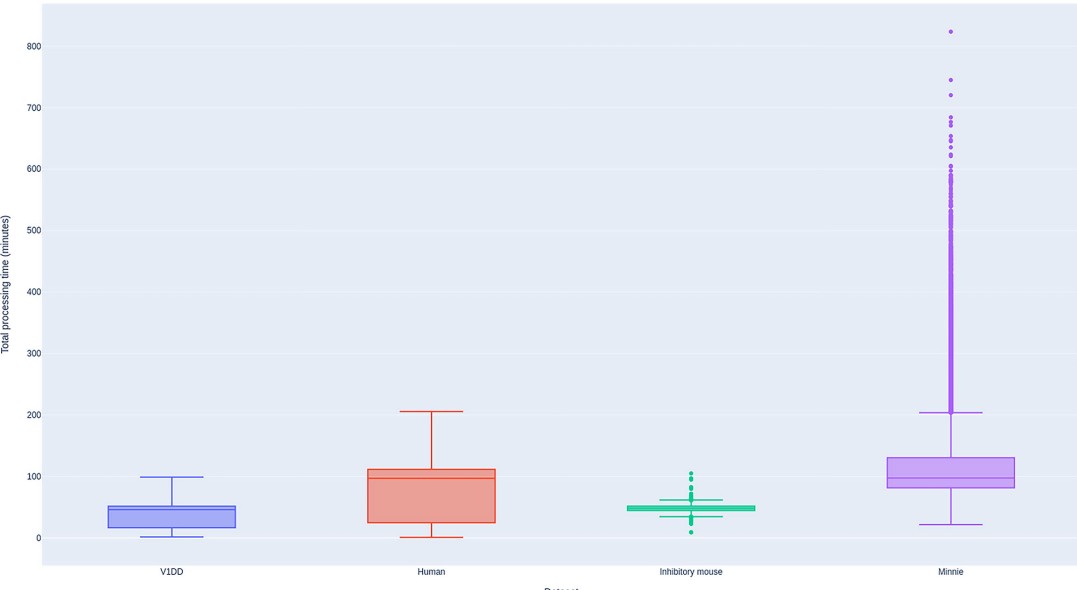

**Appendix 4—figure 1.** Performance of BlueSky workflow manager. Total processing time for sections montaged using the BlueSky workflow manager for all the datasets. Processing times shown are based on running the job in a single computing node in every job queue.

## Appendix 5

### Hardware configuration of all our services

The data transfer system (Aloha) by design scales out to multiple hosts, and networking is often the key factor for throughput. Currently, it is running (asymmetrically) on one of the following configurations:

- 2× Intel Xeon Platinum 8160, 1 TB RAM
- 2× Intel Xeon E5-2630 v4, 128 GB RAM

BlueSky is a lightweight service and shares hardware resources with the Render service and Mongo database. The hardware configuration is as follows:

- 2× Intel Xeon Gold 6128 CPU, 512 GB RAM, 6x HPE 12 TB SAS 7200 RPM HDD

The TEM database is also a lightweight service except that it requires a Mongo database, but the service rests on a dedicated server with the following configuration:

- 2× Intel Xeon E5-2630 v4, 128 GB RAM, 2 × 1 TB Seagate pro SSDs.

