## [Editor Report]

Datasets in volume electron microscopy have been growing fruit of the labor of the combined efforts of sample preparation specialists and electron microscopy engineers. A missing piece has been a method for the automation of the composition of continuous volumes out of collections of individual image tiles capable of handling the growing scales of the datasets. Pushing the boundaries of what is possible, this work illustrates how a successful approach looks like, demonstrated by its application to cubic millimeter volumes imaged at nanometer resolution. All being said, this work is but step 1 of a two-step process, whereby first a coarse but mostly correct alignment is computed, and then a refinement step using more local cues and with existing methods is applied, setting the stage for the subsequent automated reconstruction of neuronal arbors and their synapses from which to infer a cellular connectome.

---

## [Decision Letter]

**Decision letter after peer review:**

Thank you for submitting your article "A Scalable and Modular Automated Pipeline for Stitching of Large Electron Microscopy Datasets" for consideration by *eLife*. Your article has been reviewed by 3 peer reviewers, including Albert Cardona as Reviewing Editor and Reviewer #1, and the evaluation has been overseen by Ronald Calabrese as the Senior Editor. The following individuals involved in review of your submission have agreed to reveal their identity: Ignacio Arganda-Carreras (Reviewer #2); Christian Tischer (Reviewer #3).

Essential revisions:

1) Please make more clear and visible that the current framework, while impressively handling enormously large collections of image tiles, requires additional fine alignment for use in e.g., cellular connectomics.

2) Please could you elaborate on how the achieved registration accuracy relates to what would be needed to solve the scientific task.

*Reviewer #2 (Recommendations for the authors):*

In general, the paper is easy to read and the main ideas are easy to follow. However, some sections seem too technical for a non-expert audience and might benefit from adding some definitions (as a glossary or maybe footnotes). For example, in the "Software infrastructure supporting stitching and alignment" section: "It includes REST APIs for clients to GET/POST images […]", "[…] backed by a MongoDB document store that contains JSON tile specifications […]"

Regarding reproducibility, although the effort on providing open source solutions is remarkable, some documentation on how to set up the full pipeline would be greatly appreciated by the community, especially for a more modest or toy example. So far, the main site of the project (https://github.com/AllenInstitute/asap-modules) reads "We are not currently supporting this code, but simply releasing it to the community AS IS but are not able to provide any guarantees of support, as it is under active development. The community is welcome to submit issues, but you should not expect an active response.". While understandable, this is discouraging for all the potential users and it lessens the impact of the paper.

Nothing is mentioned about the specifics of the hardware resources needed to process each of the datasets. Indeed, it is not clear if the same exact computing power was used in all cases. Including those details would help the readers have a better idea of the scale of the problem being addressed and also of its requirements.

In Figure 2, maybe a shaded box including "ASAP" (or ASAP modules) should appear to clarify which components belong to it.

In Table 1, the total size of the datasets without overlap is a bit confusing. Does it involve full stitching/registration or is it simply the size without repeated pixels?

In Figures 4, 5 and 7, some of the fonts are really small and hard to read, please enlarge them.

In Figure 6c, adding an arrow to the blood vessel text would help the non-expert eyes.

In Table 2, it would be very interesting to see an estimation of the time spent in proofreading and quality control tasks, thus showing the degree of automation of the process. Also, in that caption it reads " The stitching time for all the datasets include" where it should read " The stitching time for all the datasets includes".

On page 16, where it reads "A description of these tools are as follows;" the sentence should end with a colon.

*Reviewer #3 (Recommendations for the authors):*

Software

1. ASAP-modules: The documentation is missing. It looks like the CI-generated docpages still refer to "render-modules", which seems to be deprecated/old.

2. The installation is not straight-forward. It requires adding deprecated Qt4 (700MB of extra downloads) libraries to support building an old (>3 years) opencv dependency, which is a lengthy procedure. Many dependencies are not available through the standard installation procedure (python setup or pip) but require manual intervention or actively downgrading installed packages using conda. It would be great if the installation could be simplified.

3. em_stitch: We could not find documentation or installation/usage tutorials for this software. The manuscript introduces this software as a stand-alone version of the alignment workflow, an interested reader will therefore likely start by exploring this software. Thus, an accessible tutorial including sample data will be very helpful. If em-stitch does not offer the full functionality to align "small" tiled datasets, a tutorial for setting up a simple pipeline (using the applicable tools) for a small example dataset would be very useful to someone interested in implementing the pipeline.

General remarks regarding the manuscript

1. The order in which the pipeline is presented is slightly confusing as it does not fit with the logical progress of tasks. While PointMatch determination and quality of the Solve step is shown in Figure 4, the QC determined by PM and resulting analysis in acquisition quality is shown afterwards in Figure 5.

2. What is the data resulting from running the pipeline? We assume it is a cloud-stored volume in N5 format to be visualised using Neuroglancer, but this does not become clear in the article. Does the pipeline also support outputs in different formats? (Sub-region as Tiff stack for local analysis in "traditional" software, OME-Zarr,..)

3. In line with our suggestion in the "Public Review" we suggest expanding Figure 1a such that the individual images are visible, e.g. such that one could appreciate the lens corrections, and include screenshots such as presented in Figure 3 and 6, as well as conceptual drawings such as in Figure 5a. The content of Figure 1b could then come later in the article, e.g, in a section on the "software stack and implementation details". Essentially separating example data and concepts (new Figure 1) from implementation details (Figure X). We also suggest just showing one of the options in Figure 1a(dddd) to have more space for the aforementioned additions from the other figures. Like this we would hope that the first part of the publication could provide an attractive overview of the concepts as well as some examples for a broader audience.

Detail questions regarding the manuscript:

1. QC: The metrics used for quality control are not entirely clear. Is it purely the number of PointMatches and their deviation for each tile pair?

2. What is a typical manual intervention for tackling miss-alignments? How is it done in practice? How exactly does the mentioned "parameter optimization" work?

3. Figure 5b-e: The actual shape of the distributions is not discussed in the text, thus a table with median, min, max may be sufficient for the main text and the distributions could go to the supplement. Would it be possible to give the residuals (also) in nm instead of pixels? This would make it easier to judge whether the accuracy is sufficient for connectomics.

4. Figure 6: Is this a rough or fine alignment (compared with Suppl. Figures1-3)? If it depicts only the rough aligned data, please provide an idea of how the final result looks like.

---

## [Author Response]

Essential revisions:1) Please make more clear and visible that the current framework, while impressively handling enormously large collections of image tiles, requires additional fine alignment for use in e.g., cellular connectomics.

We have made more clear throughout the text that in the presented datasets we used SEAMLess for fine alignment:

In the “Development of a stitching and alignment pipeline” of the Results section

“For all the datasets presented in this manuscript, the 2D stitching and global alignment was performed using ASAP, and afterwards the data was materialized and transferred outside of ASAP for fine alignment using SEAMLESS [18].”

In figure 2:

“The infrastructure permits multiple possible strategies for 3D alignment, including a chunk based approach in case it is not possible to 3D align the complete dataset at once, as well as using other workflows outside ASAP [18] (https://www.microns-explorer.org/cortical-mm3) for fine 3D alignment with the global 3D aligned volume obtained using ASAP.”

In the “Performance of the volume assembly pipeline” of the Results section:

“For the datasets described in this manuscript, this global alignment was the initialization point for the fine alignment done outside ASAP with SEAMLess [18]. The infrastructure present in ASAP can be however extended to "fine" alignments because ASAP is ready to implement 3D transformation both at the level of sections and at the level of individual image tiles. The quality of the fine alignment will depend on the transform that the user chooses to implement, ASAP is just a framework/vehicle for that transform.”

This topic is also addressed in the reply to reviewers 1 and 3 comments (R1-C7, R1-C9 and R3-C10) below which we also copy here.

2) Please could you elaborate on how the achieved registration accuracy relates to what would be needed to solve the scientific task.

One of our major scientific tasks in creating and analyzing these datasets is the analysis of connectivity between neurons. A major source of this connectivity is chemical synapses, and in the neocortex the most common location of synapses are dendritic spines. Spines are formed by a head (where the synapse is located) and a thin neck that connects the head to the main dendritic shaft. We reason that these small processes would be some of the most sensitive structures to misalignments of the imaging tiles and therefore we aimed for an accuracy of the stitching that would be approximately 10 times higher than the average diameter of a spine neck. The spine neck diameter is usually on average ~200 nm but has a broad range across different spines, hence the choice of 5 pixels (~ 20 nm). Twenty nanometers is also half the diameter of many of the very thin spine necks in the volume.

Bellow we also address individually each of the reviewers comments and recommendations.

Reviewer #2 (Recommendations for the authors):In general, the paper is easy to read and the main ideas are easy to follow. However, some sections seem too technical for a non-expert audience and might benefit from adding some definitions (as a glossary or maybe footnotes). For example, in the "Software infrastructure supporting stitching and alignment" section: "It includes REST APIs for clients to GET/POST images […]", "[…] backed by a MongoDB document store that contains JSON tile specifications […]"

We thank the reviewer for the suggestion. We have added a Glossary section to the manuscript that includes descriptions of the technical acronyms in the manuscript.

Regarding reproducibility, although the effort on providing open source solutions is remarkable, some documentation on how to set up the full pipeline would be greatly appreciated by the community, especially for a more modest or toy example. So far, the main site of the project (https://github.com/AllenInstitute/asap-modules) reads "We are not currently supporting this code, but simply releasing it to the community AS IS but are not able to provide any guarantees of support, as it is under active development. The community is welcome to submit issues, but you should not expect an active response.". While understandable, this is discouraging for all the potential users and it lessens the impact of the paper.

We have updated our github repository to include the most recent version of the documentation for ASAP and its modules. The documentation now includes complete step by step instructions to install and run ASAP and its modules. We do provide support for users wishing to use our software and also welcome their suggestions and improvements in the form of pull requests and active discussions as we have done in the past.

Nothing is mentioned about the specifics of the hardware resources needed to process each of the datasets. Indeed, it is not clear if the same exact computing power was used in all cases. Including those details would help the readers have a better idea of the scale of the problem being addressed and also of its requirements.

We have included descriptions and details of our hardware resources that were used in the volume assembly process. These details are included in the Methods section “BlueSky workflow engine for automated processing”. The following paragraph included in the manuscript details this information.

“For all our datasets, BlueSky utilized three different kinds of hardware nodes in our HPC cluster. The configurations are as follows;

– 2x Intel xeon 2620 processor with 256GB RAM

– 2x Intel xeon 2630 processor with 256GB RAM

– 2x Intel xeon gold 6238 processor with 512GB RAM

All our datasets were processed using a maximum of 50 nodes to perform 2D stitching of sections in parallel. A combination of the nodes with the above configuration was used in processing.”

In addition to this, the performance of the workflow manager (BlueSky) that utilizes these resources for processing our datasets is illustrated in Supplemental figures 7 and 8.

In Figure 2, maybe a shaded box including "ASAP" (or ASAP modules) should appear to clarify which components belong to it.

ASAP modules are loaded in our HPC compute cluster during processing and it interacts with other components of the workflow. We have modified Figure 3 (previous Figure 2) to include this change.

In Table 1, the total size of the datasets without overlap is a bit confusing. Does it involve full stitching/registration or is it simply the size without repeated pixels?

This is the size without repeated pixels. We have added to the manuscript:

“The details about these datasets are shown in Table 1, where the ROI size and total non-overlapping dataset size (without repeated pixels) were determined from montage metadata including pixel size and nominal overlap.”

In Figures 4, 5 and 7, some of the fonts are really small and hard to read, please enlarge them.

The fonts on all the suggested figures were increased. Please note that the figure numbers have been increased by 1 due to the addition of a new Figure 1 with the introduction to the pipeline, following the suggestion of reviewer 3.

In Figure 6c, adding an arrow to the blood vessel text would help the non-expert eyes.

We have made appropriate changes to the figure to make it more clear for the readers. Please note that the figure numbers have been increased by 1 due to the addition of a new Figure 1 with the introduction to the pipeline, following the suggestion of reviewer 3.

In Table 2, it would be very interesting to see an estimation of the time spent in proofreading and quality control tasks, thus showing the degree of automation of the process. Also, in that caption it reads " The stitching time for all the datasets include" where it should read " The stitching time for all the datasets includes".

The ASAP processing times also include the manual QC processing time duration. For each dataset, the manual QC processing time is roughly a few minutes per serial section, this was however not quantified thoroughly for the sake of reporting and therefore we do not report it in the manuscript.

On page 16, where it reads "A description of these tools are as follows;" the sentence should end with a colon.

We thank the reviewer for the careful review. We have made appropriate changes to the punctuations in the text.

Reviewer #3 (Recommendations for the authors):Software1. ASAP-modules: The documentation is missing. It looks like the CI-generated docpages still refer to "render-modules", which seems to be deprecated/old.

The documentation to ASAP-modules has been updated to match with the latest version of the code. The github repository now points to the correct and detailed version of the documentation.

2. The installation is not straight-forward. It requires adding deprecated Qt4 (700MB of extra downloads) libraries to support building an old (>3 years) opencv dependency, which is a lengthy procedure. Many dependencies are not available through the standard installation procedure (python setup or pip) but require manual intervention or actively downgrading installed packages using conda. It would be great if the installation could be simplified.

We are very thankful to the Reviewer, for going through the installation process. We have implemented the suggestions and fixes. Specifically, we added a new pypi publishing workflow to em_stitch, render-python, and bigfeta. With these modifications we’ve been able to update the stack to expect modern versions of opencv. Now ASAP will also install with pip and run tests in a fresh docker container. We’ve always been quick to help when others trying to use the software have issues and we changed the language on the “Level of Support '' which was also a suggestion from Reviewer 2.

3. em_stitch: We could not find documentation or installation/usage tutorials for this software. The manuscript introduces this software as a stand-alone version of the alignment workflow, an interested reader will therefore likely start by exploring this software. Thus, an accessible tutorial including sample data will be very helpful. If em-stitch does not offer the full functionality to align "small" tiled datasets, a tutorial for setting up a simple pipeline (using the applicable tools) for a small example dataset would be very useful to someone interested in implementing the pipeline.

We simplified the installation process for em_stitch and added these instructions to the repository's README file. We also added two jupyter notebooks which can be used with the data in the public repository -- one which derives a lens correction and shows an example plotting the distortion field, and another which solves for a montage given the derived lens correction. Both of these examples use point correspondences that were collected by the autoTEM at the time of acquisition, but can be implemented using any method for defining point correspondences.

General remarks regarding the manuscript1. The order in which the pipeline is presented is slightly confusing as it does not fit with the logical progress of tasks. While PointMatch determination and quality of the Solve step is shown in Figure 4, the QC determined by PM and resulting analysis in acquisition quality is shown afterwards in Figure 5.

Figures 4 and 5 in the current submission shows the QC plots and QC statistics plots while Figure 6 (previous Figure 5 in the original submission) shows the stitching quality not the acquisition quality. The violin plots show the distribution of the median residual values and the scale distribution of the tile images post stitching. These two metrics are used to assess the quality of stitched montages for all our datasets.

2. What is the data resulting from running the pipeline? We assume it is a cloud-stored volume in N5 format to be visualised using Neuroglancer, but this does not become clear in the article. Does the pipeline also support outputs in different formats? (Sub-region as Tiff stack for local analysis in "traditional" software, OME-Zarr,..)

The pipeline also supports other output formats (though not explicitly Tiff). We have added to the Materialization subsection of the manuscript:

“Alignment through BigFeta produces tile metadata that can be interpreted by Render and its clients to produce "materialized" representations of the stack. These representations are rendered client-side, having transformations applied and flattening overlapping tiles. The output of materialization can be in a number of file formats including n5 and the CATMAID large data tilesource, which are included in Render. It is also possible to implement custom output formats based on available python libraries using render-python to access Render's client scripts. As SEAMLeSS expects data in the cloud-volume compatible neuroglancer precomputed format, the datasets in Table 1 were materialized with a python script that uses render-python to write to neuroglancer precomputed format using the cloud-volume library.”

3. In line with our suggestion in the "Public Review" we suggest expanding Figure 1a such that the individual images are visible, e.g. such that one could appreciate the lens corrections, and include screenshots such as presented in Figure 3 and 6, as well as conceptual drawings such as in Figure 5a. The content of Figure 1b could then come later in the article, e.g, in a section on the "software stack and implementation details". Essentially separating example data and concepts (new Figure 1) from implementation details (Figure X). We also suggest just showing one of the options in Figure 1a(dddd) to have more space for the aforementioned additions from the other figures. Like this we would hope that the first part of the publication could provide an attractive overview of the concepts as well as some examples for a broader audience.

We have added a new Figure 1 addressing most of the reviewer suggestions, including the broad overview, conceptual drawings and as well more detail of individual images. Figure 1 of the initial submission is now Figure 2.

Detail questions regarding the manuscript:1. QC: The metrics used for quality control are not entirely clear. Is it purely the number of PointMatches and their deviation for each tile pair?

To clarify this point we changed the manuscript text to:

“The stitching issues that are identified include misalignments between stitched tiles, gaps in montages, and seams. These issues are identified based on the mean residual between point correspondences from every pair of tiles. This represents how well the point correspondences have aligned from each of the tiles after montage transformations are applied to them (see Figure 4.a). This metric is represented in pixel distance and is used to locate areas of misalignments and seams. The gaps in stitching are identified by means of how many neighbors a tile image has before and after stitching and based on their area of overlap with its neighbors. A seam appears as misalignment between a tile and many of its neighbors and is identified using a cluster of point correspondences whose residuals are above a certain threshold. In addition to these metrics, we also compute the mean absolute deviation (MAD) (see Figure 5) that measures the amount of distortion a tile image undergoes with the transformation. The MAD statistics is a measure using which we identify montages that are distorted (see Figure 5) once it passes the automated QC identifying other issues. Since the crux of our computations are based on the point correspondences, we also generate plots to quickly visualize the density of point correspondences between tile images within a section (see Figure 4.k).”

2. What is a typical manual intervention for tackling miss-alignments? How is it done in practice? How exactly does the mentioned "parameter optimization" work?

To address this question as well as the comments R1C6 from Reviewer 1 we have made added the following text as well as a new Supplementary Figure 1:

“Sections that failed QC are examined by a human proofreader and moved to the appropriate stage of re-processing. A manual proofreading process typically includes examining the QC plot for issues and further visualizing those areas in montage with issues to ensure that those issues either correspond to resin or film region tiles or tiles corresponding to the tissue region. The regions with misalignments are further examined to send them to the appropriate stage of processing. If the misalignments are caused due to insufficient point correspondences, then they are sent to the point-matching stage of the montage workflow for generation of point correspondences at a higher resolution. Misaligned sections with sufficient point correspondences are sent to the solver stage with new parameters. These parameters were heuristically chosen by means of a parameter optimization algorithm based on the stitching quality metrics (see Methods section for more details and Supplementary Figure 1 for optimized parameter selection plots).”

3. Figure 5b-e: The actual shape of the distributions is not discussed in the text, thus a table with median, min, max may be sufficient for the main text and the distributions could go to the supplement. Would it be possible to give the residuals (also) in nm instead of pixels? This would make it easier to judge whether the accuracy is sufficient for connectomics.

We have included a table showing the median residual values and median x and y scale values in a table (Table 2) in addition to the description of the Figure 5b-e (now Figure 6b-e). The violin plots in the figure depicts the density distribution of the median residual values computed for every serial section from our datasets and are grouped by the acquisition systems. It can be seen that the density of distribution is below the threshold value (the horizontal line in these plots) indicating the stitching quality of the serial sections.

4. Figure 6: Is this a rough or fine alignment (compared with Suppl. Figures1-3)? If it depicts only the rough aligned data, please provide an idea of how the final result looks like.

Figure 7 (Figure 6 in the original submission) as well as supplemental figures 2-5, show the cross sectional views of the global alignment (“rough” alignment) of the four datasets discussed in this work. The final result after fine alignment with SeamLESS can be found in www.microns-explorer.org/mm3. Please see also reply to R1C7